# Single-cell mRNA profiling reveals the hierarchical response of miRNA targets to miRNA induction

Andrzej J Rzepiela[1,†], Souvik Ghosh[1] (iD), Jeremie Breda[1], Arnau Vina-Vilaseca[1] (iD), Afzal P Syed[1], Andreas J Gruber[1], Katja Eschbach[2], Christian Beisel[2], Erik van Nimwegen[1] & Mihaela Zavolan[1,*] (iD)

## Abstract

miRNAs are small RNAs that regulate gene expression post-transcriptionally. By repressing the translation and promoting the degradation of target mRNAs, miRNAs may reduce the cell-to-cell variability in protein expression, induce correlations between target expression levels, and provide a layer through which targets can influence each other's expression as "competing RNAs" (ceRNAs). However, experimental evidence for these behaviors is limited. Combining mathematical modeling with RNA sequencing of individual human embryonic kidney cells in which the expression of two distinct miRNAs was induced over a wide range, we have inferred parameters describing the response of hundreds of miRNA targets to miRNA induction. Individual targets have widely different response dynamics, and only a small proportion of predicted targets exhibit high sensitivity to miRNA induction. Our data reveal for the first time the response parameters of the entire network of endogenous miRNA targets to miRNA induction, demonstrating that miRNAs correlate target expression and at the same time increase the variability in expression of individual targets across cells. The approach is generalizable to other miRNAs and post-transcriptional regulators to improve the understanding of gene expression dynamics in individual cell types.

**Keywords** ceRNA; Michaelis–Menten constant; miRNA regulation; scRNA-Seq; target down-regulation

**Subject Categories** Quantitative Biology & Dynamical Systems; RNA Biology

**Mol Syst Biol. (2018) 14: e8266**

## Introduction

miRNAs guide Argonaute proteins to mRNA targets, repressing their expression post-transcriptionally (Huntzinger & Izaurralde, 2011). Measurements of transcript and protein levels following perturbations in the levels of individual miRNAs showed that the fundamental molecular mechanism of mammalian miRNAs is target destabilization, through the recruitment of factors that promote mRNA decay (Lim *et al*, 2005; Hausser *et al*, 2009; Guo *et al*, 2010; Bartel, 2009). However, time series of mRNA and protein-level measurements after miRNA transfection also revealed that repression of target translation precedes the increase in its degradation rate (Bazzini *et al*, 2012; Hausser *et al*, 2013; Eichhorn *et al*, 2014). A miRNA typically has hundreds of evolutionarily conserved target sites (Lewis *et al*, 2005; Grün *et al*, 2005; Gaidatzis *et al*, 2007), yet only very few predicted targets are down-regulated more than twofold in miRNA transfection experiments (Hausser & Zavolan, 2014). Whereas disruption of miRNA biogenesis impairs the ability of embryonic stem cells to differentiate (Kanellopoulou *et al*, 2005), and some miRNAs such as the founders of the class, the lin-4 and let-7 miRNA of *Caenorhabditis elegans* have striking developmental phenotypes (Ha *et al*, 1996; Wightman *et al*, 1993; Reinhart *et al*, 2000), most miRNA genes are individually dispensable for development and viability, at least in the worm (Miska *et al*, 2007). These observations suggested that strong repression may not be the primary function of miRNAs and that other functions should be investigated (Ebert & Sharp, 2012).

A computational study of small RNA-dependent gene regulation in bacteria initially proposed that post-transcriptional regulators impose thresholds on the protein levels of their targets in response to transcriptional induction, conferring robustness to transcriptional noise (Levine *et al*, 2007). Experiments with target reporters in mammalian systems demonstrated that miRNAs could play a similar role (Mukherji *et al*, 2011). Gene expression being a stochastic process, the number of protein molecules expressed from a given gene varies between cells in a cell population. The ratio of variance to mean of the number of protein molecules per cell (the "noise" in protein expression) is predicted to be proportional to the ratio of mRNA translation and mRNA degradation rates (Shahrezaei & Swain, 2008). Intriguingly, these are the rates that miRNAs modulate so as to decrease protein expression noise. Indeed, a recent study reported increased variability in CD69 protein expression across miRNA-deficient, developing mouse thymocytes (Blevins *et al*, 2015). However, as the reduction in target protein noise is predicted to scale as the square root of the miRNA-induced change in protein level (Osella *et al*, 2011; Schmiedel *et al*, 2015), which is small for the vast majority of evolutionarily conserved miRNA targets (Hausser *et al*, 2013; Eichhorn *et al*, 2014; Hausser &

---

1 Biozentrum, University of Basel and Swiss Institute of Bioinformatics, Basel, Switzerland
2 Department of Biosystems Science and Engineering, ETH Zürich, Basel, Switzerland
*Corresponding author. Tel: +41 61 207 1577; E-mail: mihaela.zavolan@unibas.ch
†Present address: Scientific Center for Optical and Electron Microscopy, ETH Zürich, Zürich, Switzerland

Zavolan, 2014), it is unlikely that many of the predicted miRNA targets are regulated in this manner.

It has also been proposed that at the cellular level, miRNAs provide a "channel" through which the many predicted miRNA targets "communicate" as "competing RNAs" (ceRNAs; Poliseno *et al*, 2010; Figliuzzi *et al*, 2013; Cesana *et al*, 2011; Karreth *et al*, 2015; Wang *et al*, 2013). Rough estimates of the number of potential binding sites for a miRNA (also called miRNA "target abundance") are in the range of $\sim 10^5$ sites per cell, much higher than the number of cognate miRNA molecules (Denzler *et al*, 2014). In this regime, where the targets are already in high excess relative to the miRNAs, overexpressing a single target could not appreciably affect the expression of the other targets. Yet, examples of ceRNAs continue to emerge (Poliseno *et al*, 2010; Figliuzzi *et al*, 2013; Cesana *et al*, 2011; Karreth *et al*, 2015; Wang *et al*, 2013). These estimates of target abundance did not consider the possibility that targets may not be equivalent in their ability to bind and sequester miRNAs. Indeed, a computational analysis suggested that miRNA targets have asymmetric relationships, high-affinity targets being able to sequester miRNAs from low-affinity targets, at comparable target concentrations, but not the other way around (Figliuzzi *et al*, 2013). *In vitro* measurements indicate that miRNA target sites can have widely different affinities for the miRNA–Argonaute complex (Wee *et al*, 2012), an observation that is supported by measurements of Argonaute-dwelling times on individual miRNA target sites (Chandradoss *et al*, 2015). However, estimates of *in vivo* miRNA–target interaction constants are lacking.

Taking advantage of a system in which the expression of a single miRNA precursor can be induced over a wide concentration range, we measured the transcriptomes of thousands of individual cells and assessed how the expression levels of miRNA targets relate to the expression level of the miRNA. We obtained experimental evidence for behaviors that were previously suggested by computational models or evaluated only with miRNA target reporters. These include the non-linear, ultrasensitive response of miRNA targets to changes in the miRNA concentration as well as the dependency of the variability in target levels between cells on the concentration of the miRNA. Furthermore, we found that only a small fraction of predicted targets are highly sensitive to changes in miRNA expression. With a computational model, we illustrate how these targets can influence the expression of other targets as competing RNAs. Our approach is applicable to other post-transcriptional regulators of mRNA stability, allowing the analysis of their concentration-dependent impact on the transcriptome.

## Results

### A system to study the impact of miRNA expression on the transcriptome of individual cells

miRNA target reporters are widely used to study miRNA-dependent gene regulation. However, these reporters are often expressed at much higher levels than when expressed from their corresponding genomic loci. Furthermore, these reporters do not respond to the regulatory influences to which the endogenous transcripts respond. To circumvent these issues and investigate the crosstalk of miRNA targets in their native expression context, we

used a human embryonic kidney (HEK) 293 cell line, i199 (Hausser *et al*, 2013), in which the expression of the hsa-miR-199a miRNA precursor and of the green fluorescent protein (GFP) can be simultaneously induced by doxycycline, from a pRTS1 episomal vector (Fig 1A). To assess the reproducibility of the inferred sensitivity parameters for miRNA targets, we used a related cell line, i199-KTN1 (Hausser *et al*, 2013), derived from i199 through the stable integration of a target of hsa-miR-199a-3p. This target consisted of the Renilla luciferase coding region followed by the 3′ untranslated region (UTR) of kinectin 1 (KTN1). We reasoned that these similar but not identical cell lines should allow us to assess the reproducibility of the inferred parameters, which we do expect to vary between more distant cell types due to differences in the expression of regulatory factors.

The processing of hsa-miR-199a gives rise to two mature miRNAs, hsa-miR-199a-5p and hsa-miR-199a-3p. These miRNAs have distinct "seed" sequences (at positions 2–7 of the miRNA) and therefore largely non-overlapping target sets; only seven of the top 100 targets predicted (Gumienny & Zavolan, 2015) for each miRNA are shared. The bidirectional nature of the promoter in the pRTS1 vector was characterized before, by fluorescence-activated cell sorting (Bornkamm *et al*, 2005). In our construct, the luciferase protein-coding sequence has been replaced by a pri-miRNA. As no method is currently available for simultaneously measuring the expression of a miRNA and of a protein-coding gene in single-cells, we assessed whether the two bi-directionally transcribed RNAs have correlated expression in cell populations. Indeed, by RT–PCR we found that the expression of hsa-miR-199a-5p and expression of the GFP mRNA, in cell populations induced with different concentrations of doxycycline, were highly correlated (Fig 1B, Spearman's $r = 0.91$, $P = 1.74\text{E-}07$). Furthermore, the expression of both mature miRNAs processed from the hsa-miR-199a precursor increased in parallel to the concentration of the inducer, as expected (Appendix Fig S1A). Altogether, these data indicate that the level of GFP mRNA can serve as a "proxy" for the miRNA levels in studying the response of miRNA targets to the miRNA in individual cells. Carrying out Argonaute 2 protein crosslinking and immunoprecipitation in fully induced (1 μg/ml doxycycline) HEK 293 cells, we confirmed that hsa-miR-199a-5p and hsa-miR-199a-3p were incorporated into the miRNA effector complex and were among the highest represented miRNAs (Appendix Fig S1B).

We then induced cells with doxycycline concentrations spanning the 0–1 μg/ml range (as described in Materials and Methods), pooled the cells and carried out mRNA 3′ end sequencing of 3,280 distinct i199 and 3,143 i199-KTN1 cells, on a 10x Genomics platform. In parallel, we carried out bulk mRNA sequencing from both non-induced i199 cells and cells that were fully induced (1 μg/ml doxycycline). The distribution of the number of distinct transcripts obtained from individual single-cells is shown in Appendix Fig S1C. GFP mRNAs were captured from 43% of the i199 cells, in which the mean GFP mRNA expression was 32 transcripts per million (TPM; Fig 1C). mRNA expression levels inferred either by averaging over single-cells (SCs) with no GFP mRNA or from bulk sequencing of non-induced cell populations (CP) were highly correlated (Spearman's $r$ of $\log_2$ expression values = 0.89, $P <$ 1E-15, Fig 1D). The expression of the top 100 MIRZA-G-C-predicted targets of the two miRNAs (Gumienny & Zavolan, 2015) was significantly lower in cells with high GFP mRNA expression (> 6.8

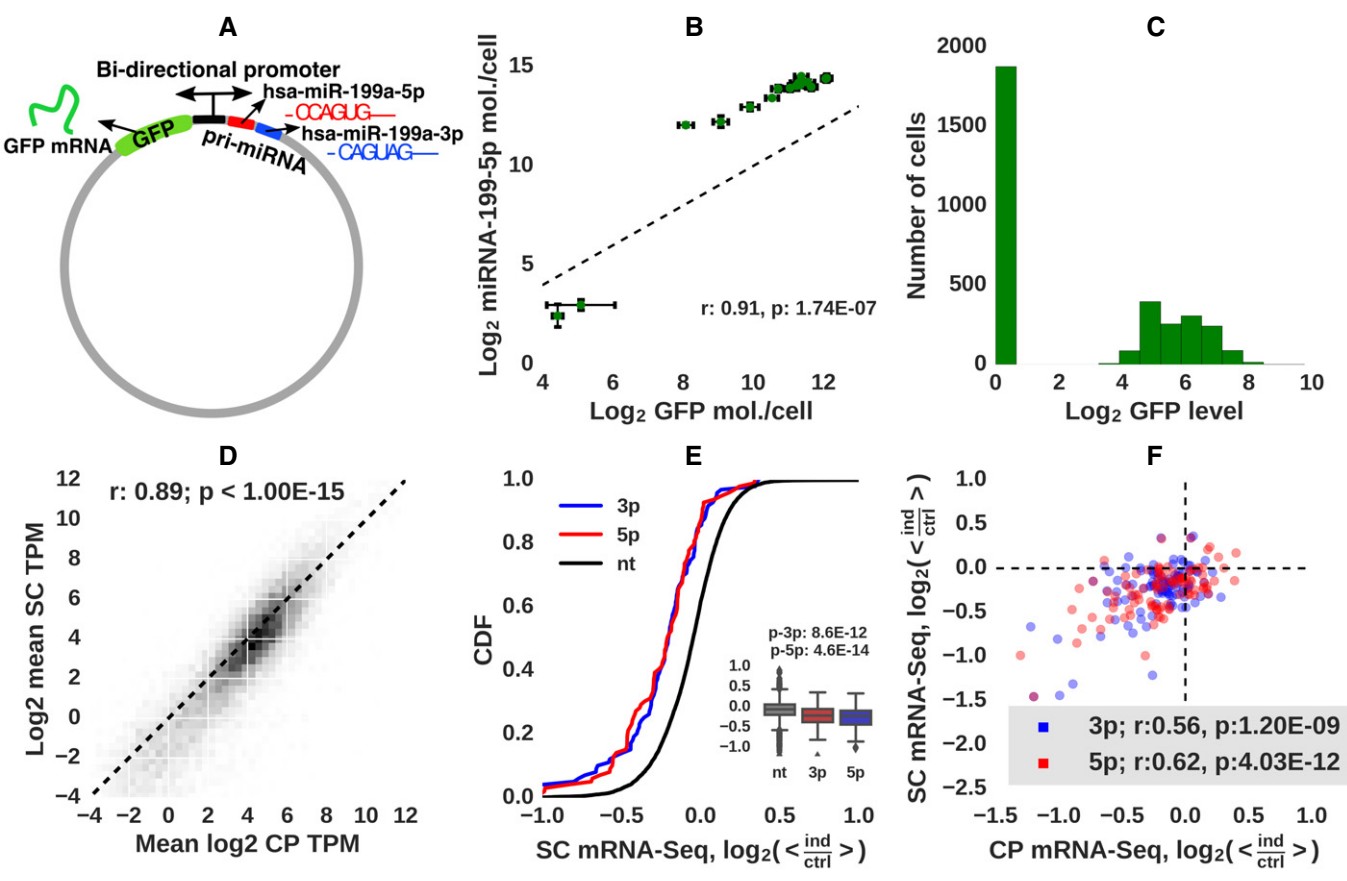

**Figure 1.  Design and characterization of the experimental system.**

A   Schematic representation of the construct used to express hsa-miR-199a-5p (red), hsa-miR-199a-3p (blue), and the reporter GFP mRNA from a bidirectional promoter. Shown are also the "seed" sequences (nucleotides 2–7) of the two miRNAs.

B   The expression levels of hsa-miR-199a-5p and GFP mRNA, measured from cell populations by quantitative PCR, are highly correlated. Error bars show standard deviations from two replicates.

C   Histogram of normalized GFP mRNA expression (TPM) in individual i199 cells.

D   Correlation of mRNA expression levels estimated from SC sequencing (1,875 $T^0$ cells (see text for definition) from which no GFP mRNA was captured) and from CP mRNA-Seq (six replicates of non-induced cell populations).

E   Cumulative distribution of expression differences of the top 100 targets of hsa-miR-199a-5p (red), of top 100 targets of hsa-miR-199a-3p (blue), and of 7,347 remaining, "background" genes (black) between cells expressing highest and lowest GFP levels [216$T^\infty$ cells with > 6.8 TPM GFP ("ind") vs. 1,875 $T^0$ cells with 0 TMP GFP ("ctrl")]. Boxplots of $\log_2$-fold change of non-targets, top 100 miR-199a-3p, and top 100 miRNA-199a-5p targets are shown in the inset. *P*-values of the rank-sum test comparing targets and non-targets are also shown. Horizontal line is a mean, box shows where 50% of data points are (interquartile range, IQR), whiskers show points within 1.5 * IQR  from 25/75-percentile border of the box.

F   Scatter plot of expression differences of the top 100 targets of each miRNA, estimated from bulk sequencing (CP) or from single-cell sequencing ($T^\infty$ and $T^0$ cells defined as for previous panel).

TPM) compared to cells with no GFP expression (0 TPM, Fig 1E). Importantly, the expression of predicted targets decreased in parallel with the increase in GFP mRNA levels, further indicating that the GFP mRNA is a good proxy for the miRNA expression in individual cells (Appendix Fig S1D and E). Finally, the miRNA-induced changes in target expression, inferred either from bulk or from single-cell sequencing of strongly induced and uninduced cells were significantly correlated (Fig 1F). The results were reproduced in the related i199-KTN1 cell line (Appendix Fig S2). Results of a parallel analysis with miRNA targets predicted by TargetScan (Garcia *et al*, 2011) are shown in Appendix Figs S6–S12. Altogether, these results indicate that the system behaves as expected and can be used for further analysis of miRNA-dependent gene regulation in single-cells.

**miRNA targets follow theoretically predicted behaviors in response to miRNA induction**

The dynamics of small networks composed of miRNAs, and targets have been investigated computationally, with stochastic models (Bosia *et al*, 2013; Figliuzzi *et al*, 2013). Bosia *et al* (2013) predicted that the coefficient of variation ($C_V$) of miRNA targets increases with the transcription rate of the miRNA, showing a local maximum in the region where the miRNA and targets are in equimolar ratio. The correlation of expression levels of mRNAs that are targeted by the same miRNA was predicted to exhibit a maximum around the same threshold. We used a similar simple model of miRNA-dependent gene regulation to predict the behavior of targets in our experimental system. Briefly, we considered *M* mRNA targets of a given miRNA,

each with a specific transcription rate $\alpha_i$, decay rate $\delta_i$, and level $m_i$, with $i \in \{1, \ldots M\}$. Target $i$ could bind a miRNA-containing Argonaute (Ago) complex at rate $k_{on_i}$ and dissociate from the complex at rate $k_{off_i}$. Because in our experimental system we induced miRNA expression to specific stable levels before carrying out the mRNA sequencing, we neglected the dynamics of the miRNA and assumed that the total number $A$ of Ago-miRNA complexes in a given cell was constant, though varying between cells. The number of free Ago-miRNA complexes is then given by $A_F = A - \sum_{j=1}^{M} A_j$. Finally, we assumed that Ago-miRNA-bound mRNAs decay at rates $k_{cat_i}$. Under this simple model (see also Hausser and Zavolan, 2014), free mRNAs ($m_i$) and miRNA-bound mRNAs ($A_{m_i}$) follow the dynamics described by the system of $2M$ differential equations

$$\frac{\partial m_i(t)}{\partial t} = \alpha_i - \delta_i m_i(t) - k_{on_i} m_i(t)\left(A - \sum_{j=1}^{M} A_{m_j}(t)\right) + k_{off_i} A_{m_i}(t)$$

$$\frac{\partial A_{m_i}(t)}{\partial t} = k_{on_i} m_i(t)\left(A - \sum_{j=1}^{M} A_{m_j}(t)\right) - k_{off_i} A_{m_i}(t) - k_{cat_i} A_{m_i}(t).$$

$$(1)$$

We carried out stochastic simulations of a system with four miRNA targets (Fig 2A), choosing parameters of target expression and interaction with the miRNA such that (i) target expression spanned a broad range, (ii) they underwent miRNA-dependent down-regulation at either low (targets **a** and **b**) or high (targets **c** and **d**) miRNA levels, and (iii) down-regulation of all targets was moderate, as generally observed in experiments. The response of individual *in silico* targets to miRNA induction is shown in Fig 2A. Figure 2B and C shows the variability of target expression between simulated cells and the pairwise correlations of target expression levels across all simulated cells, as functions of the total miRNA level. Similar to the predictions of Bosia *et al* (2013), the targets in our *in silico* system also experience destabilization, increased correlation, and increased expression noise, all within a limited range of miRNA expression, i.e. at a specific threshold. Figure 2B also shows that for each target, the coefficient of variation increases in function of miRNA expression level, as the target expression level is reduced by the miRNA, and that targets with low expression level have higher coefficients of variation compared to highly expressed targets. Furthermore, there is a noticeable spike in the coefficient of variation of each target, in the region where the target experiences a hypersensitive down-regulation in response to the miRNA (see also Appendix Fig S3A). The miRNA also induces correlated changes in its targets (Fig 2C); targets with high sensitivity to the miRNA that are repressed at low miRNA concentrations (**a** and **b** in our example) exhibit the highest correlation coefficient, and over a widest range of miRNA concentrations. However, targets that differ strongly in concentration of the miRNA that triggers their response or in the magnitude of miRNA-induced decay appear uncorrelated (**c** with respect to the others in our example).

We then turned to the experimental data. For both miRNAs and both cell lines, the total target level (see Materials and Methods for target selection) exhibited the expected threshold decrease in function of the GFP expression level, which we used as proxy for the miRNA expression (Fig 2D and G, and Appendix Fig S3D). The $C_V$ and $r_P$ values, computed as ratios to the corresponding values for a similarly sized set of non-targets, also showed the expected behaviors; namely, the coefficient of variation in total target expression

across individual cells increased with the GFP expression (Fig 2E and H, see also Appendix Fig S3B and E), while the mean pairwise correlation coefficient of target expression in individual cells peaked at an intermediate level of GFP mRNA expression (Fig 2F and I, see also Appendix Fig S3C and F). Randomizations showed that in spite of the large noise, indicated by the size of the error bars, the $C_V$ of targets remained larger than that of non-targets and the correlation of target expression larger than that of non-target expression. Thus, even though the noise of single-cell experiments is large and mRNA capture is incomplete, the experimental data follow the theoretical predictions and the simulations.

## The sensitivity of individual targets to miRNA regulation can be inferred from their expression in cells with varying miRNA level

We used the computational model described in equation (1) to derive two measures of target sensitivity to miRNA regulation. First, we derived the Michaelis–Menten-like constant (Wee *et al*, 2012) $K_{M_i} = \frac{k_{off_i} + k_{cat_i}}{k_{on_i}}$, defined as the ratio of the dissociation rate of mRNA $i$ from the miRNA-primed Argonaute protein (whether or not accompanied by Ago-catalyzed decay) and their rate of binding. We further derived the level of free Ago-miRNA complexes at which a specific target $i$ is halfway between its maximum level $T_i^0$, realized when the miRNA is not expressed, and its minimum level $T_i^\infty$, realized when the miRNA is in high excess relative to all targets. As shown in Materials and Methods, this critical concentration is given by

$$A_{Fi}^C = \frac{K_{M_i}}{\frac{T_i^0}{T_i^\infty}}.$$

We then devised a procedure for inferring these two parameters for each miRNA target from the experimental data, which has a high level noise (total target levels vary almost twofold in individual cells with similar GFP expression (Fig 2D and G), for reasons that may include the low mRNA capture rate and the imperfect coupling of miRNA and GFP mRNA levels). We used the system described in equation (1) to test procedures for analyzing noisy single-cell data such that we can infer target-specific parameters at the limit of accuracy afforded by the single-cell experiments. Relevant for the inference are the expression levels of targets in the absence of the miRNA, the expression levels when the miRNA is present at maximal concentration, and the expression levels in all cells in which the miRNA has intermediate expression. Thus, we generated *in silico* data with the computational model (Fig 3A), added noise in target levels comparable to the noise observed *in vivo* (Fig 3B), and then experimented with the selection of cells to use in the inference and with the smoothing of the target levels (Fig 3B and C) to most accurately recover the input parameters (see also Materials and Methods and Appendix Fig S4). In particular, different miRNA targets respond at different miRNA concentration, and only cells in which the miRNA concentration is in the relevant range for that target could be used for inferring the shape of the target's response. Thus, to select cells that are relevant for the inference of parameters of all targets in parallel, we examined the dependence of average target level as a function of the miRNA concentration in a cell. As even the average target level varies quite widely between cells with similar miRNA concentration (Fig 3B), we explored procedures for smoothing average expression levels as a function of miRNA expression

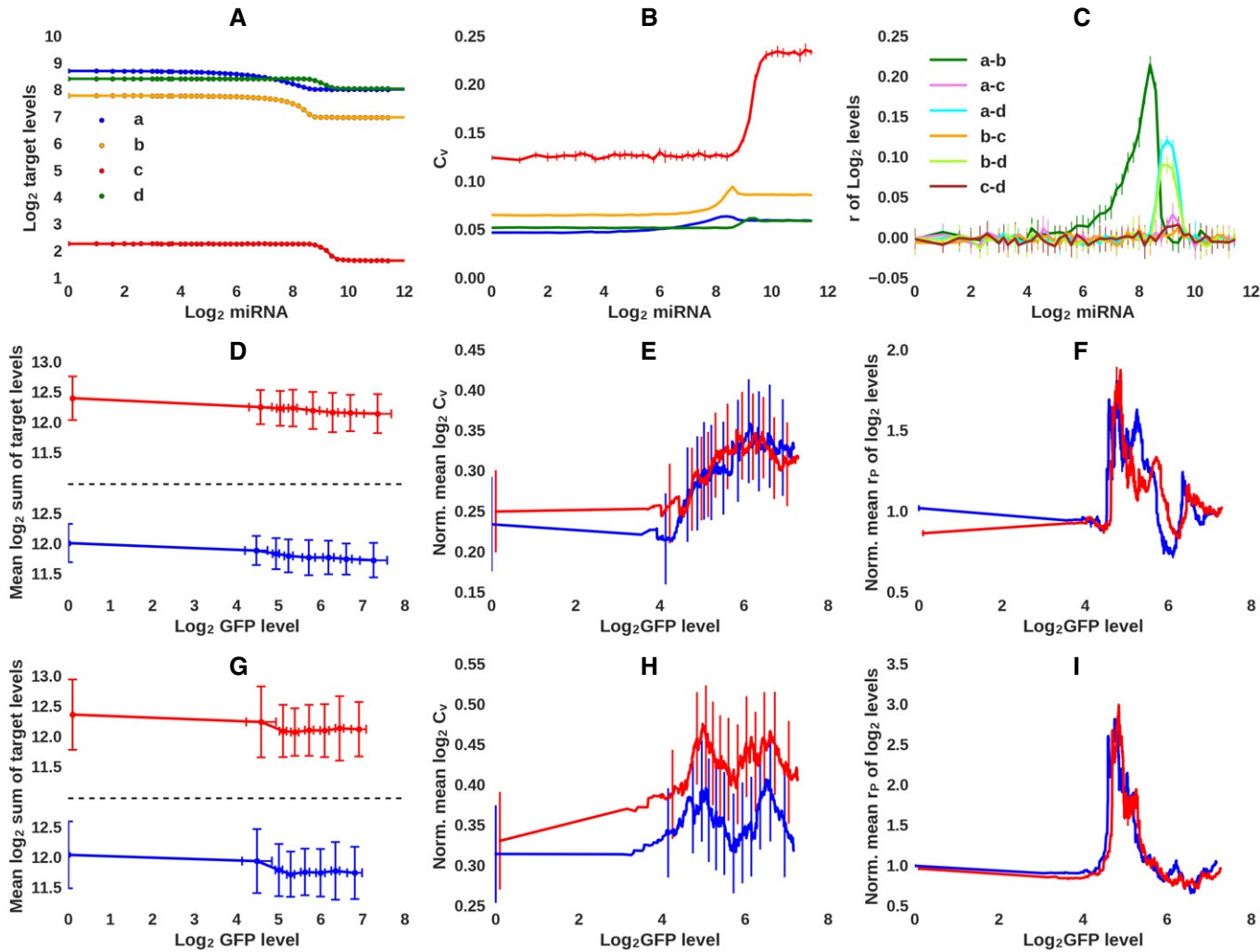

**Figure 2. Expected and observed response of miRNA targets to miRNA induction in single-cells.**

A   Results of numerical integration (equation (1), solid lines) and the average of six stochastic simulations (dots) of a model with four target genes (indicated by distinct colors) chosen to cover a wide expression range and to have either high or low sensitivity to the miRNA. Fifty *in silico* cells, each with a defined miRNA concentration, were simulated.

B   Coefficient of variation ($C_V$) of *in silico* target levels across cells, calculated in function of miRNA expression, from the simulation trajectories.

C   Pearson's correlation coefficients of expression levels of pairs of genes from *in silico* cells, calculated in function of miRNA expression from the simulation trajectories.

D–I   (D, G) Total expression (log$_2$ sum of TPM) of 100 lowest $A_F^C$ hsa-miR-199a-5p (red) and hsa-miR-199a-3p (blue) targets (see also Materials and Methods for target selection) in the i199 (D) and i199-KTN1 (G) cells, in function of log$_2$ GFP expression in the same cells. (E, H) mean $C_V$ and (F, I) mean Pearson's pairwise correlation coefficients for miRNA targets in function of GFP expression in i199 (E, F) and i199-KTN1 (H, I) cells. Averages were calculated from the 200 cells with GFP expression closest to a specific expression level. $C_V$ values are shown as ratios to corresponding values computed for all non-target mRNAs (E, H) and $r_P$ to mean of 50 evaluations of random selection of 100 non-target genes (F, I).

Data information: For (B, C, D, and G) plot, standard deviations are shown, for (E, F, H and I) plot, standard errors are shown.

before the selection of cells for the inference, as described in Materials and Methods. The region of target sensitivity to the miRNA is indicated by the red line in Fig 3B, and the gradient in mean target level as a function of miRNA concentration is shown in Fig 3C. The free miRNA levels inferred from these *in silico* data showed that only when the total miRNA level is sufficiently high to occupy all the available target sites (Fig 3D) does free miRNA accumulate, as expected. The correlation between target-specific input and recovered parameters (Fig 3E, Pearson's $r = 0.56$, $P$-value $= 3.0 \times 10^{-20}$) was at the upper bound set by the level of noise in the simulated

data, as shown by correlation between the parameters recovered from two simulations that only differed in the measurement error added to the target expression levels in the simulated cells (Fig 3F).

**A limited number of targets exhibit high sensitivity to miRNA induction**

We then turned to estimating the sensitivities of the predicted miRNA targets from the experimental data. For each miRNA, we selected the 300 MIRZA-G-C-predicted targets with highest

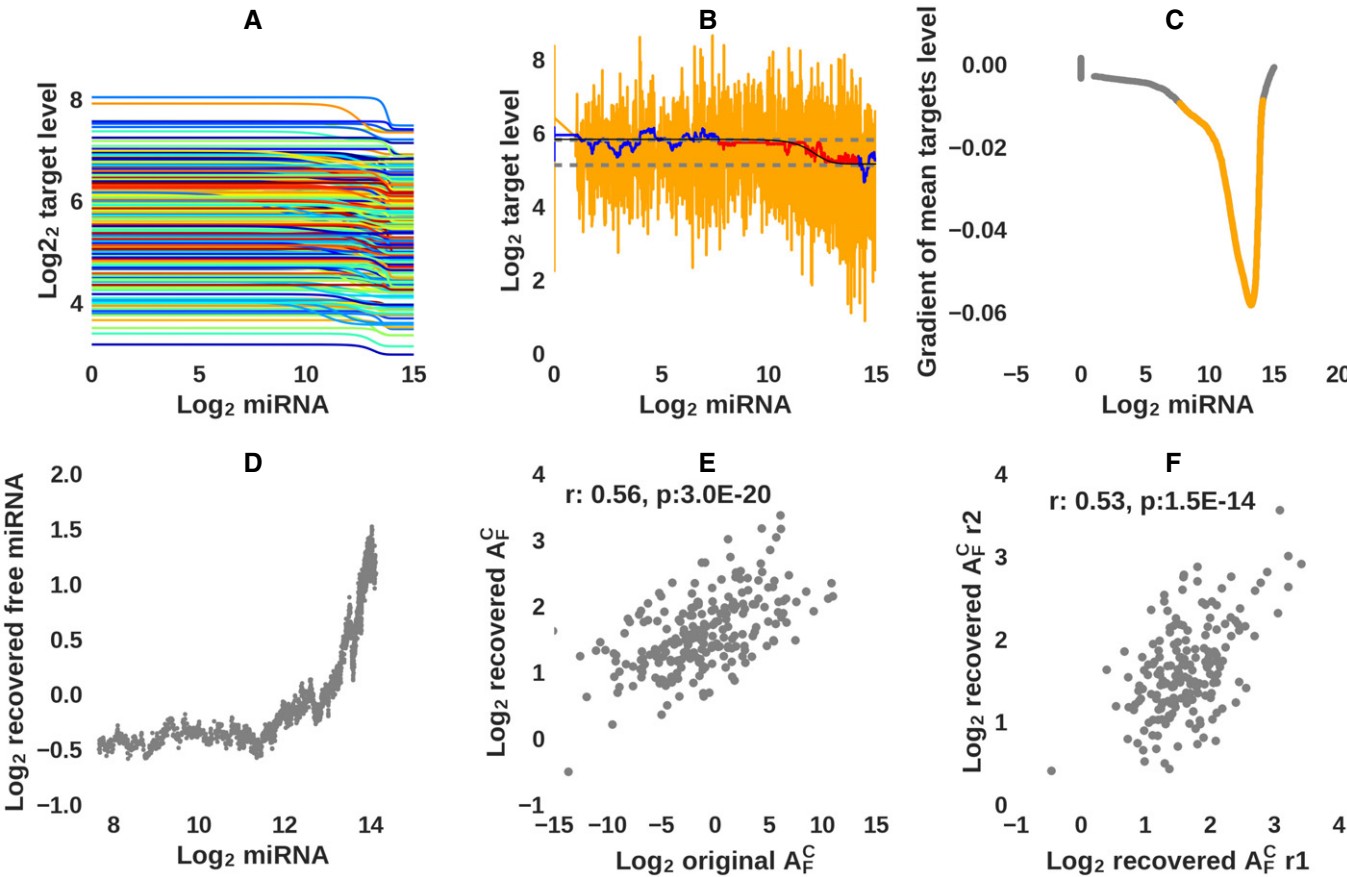

**Figure 3.  Validation of the approach for inferring target sensitivity from single-cell data.**

A   Response of 300 *in silico* targets, each with associated parameters describing its transcription, decay, rates of binding to and dissociating from the miRNA (values drawn from distributions around experimentally measured values, see Appendix Fig S4) in response to increasing miRNA concentration.

B   Noise (orange) was added to the target expression (black), and then, running means (blue) were calculated over increasingly wider windows to ensure that the estimated expression levels $T_{ij}$ for gene $i$ in cell $j$ (for cells used in the inference (red)) were between the maximum ($T_i^0$) and minimum ($T_i^\infty$) levels, corresponding to no miRNA being expressed and to the miRNA being expressed at very high levels in the cell (allowing for a small tolerance $c$; dashed lines).

C   Cells for which the gradient of the total target level with respect to the miRNA level was less than $-0.01$ (shown in orange, and corresponding to the points shown in red in panel (B)) were used to construct the $\tilde{T}$ matrix of gene expression levels per cell.

D   Scatter plot of the total miRNA levels that were used as input to the model and the levels of free miRNA inferred from the simulated data.

E   Scatter plot of the input vs. inferred $A_{Fi}^C$ values. The Pearson correlation coefficient and its associated *P*-value are also shown.

F   Scatter plot of $A_{Fi}^C$ values inferred from *in silico* data that were generated with the same input target parameters, but to which two distinct sets of "measurement errors" were applied. The Pearson correlation and associated *P*-value are also shown.

prediction scores (Gumienny & Zavolan, 2015) that had an expression level of at least ~8 TPM when the miRNA was not expressed, and underwent at least 8% down-regulation at the highest miRNA concentration ($\log_2 T_i^\infty / T_i^0 < -0.12$). We used cells with $\log_2$ GFP expression of 0 TPM (1,875 and 1,629 cells for i199 and i199-KTN1 cells, respectively) to infer target levels $T_i^0$, when the miRNA is not expressed, cells with more than 6.8 TPM GFP (216 cells for i199 and 205 for i199-KTN1) to infer target levels $T_i^\infty$ at saturating miRNA concentration, and all other cells to construct the $\tilde{T}$ matrix of individual target expression levels in single-cells with intermediate miRNA expression. Applying the inference described in Materials and Methods, we obtained $A_F^C$ (Fig 4A) and $K_M$ (Fig 4B) parameters for all targets and found that their distributions covered a fourfold to eightfold range. The average response of the 20 targets with lowest and highest values of these two

parameters to miRNA induction is shown in Fig 4C. For both hsa-miR-199a-5p and hsa-miR-199a-3p miRNAs, target parameters inferred independently from the two cell lines were significantly correlated (Fig 4D and E), indicating the robustness of our results. For hsa-miR-199a-3p, Pearson's correlation coefficients were 0.49 (*P*-value = $2.6 \times 10^{-11}$) for $A_F^C$ and 0.4 (*P*-value = $5.9 \times 10^{-8}$) for $K_M$, while for hsa-miR-199a-5p, they were 0.43 (*P*-value = $3.6 \times 10^{-8}$) for $A_F^C$ and 0.34 (*P*-value = $2.2 \times 10^{-5}$) for $K_M$. Especially apparent on the scatter plot of $A_F^C$ values is a small group of targets that respond at low miRNA concentrations and thus have low $A_F^C$ in both cell lines (see also Fig 4C). These low $A_F^C$ targets have higher prediction scores and are enriched in DNA-binding factors compared to the high $A_F^C$ targets (Appendix Tables S1 and S2, and Fig S5A). The measure that is most broadly used to validate computational target predictions is the change in expression

that predicted targets experience upon strong miRNA induction (Fig 4F). Sorting targets by their MIRZA-G-C scores and computing the average fold change [between cells with high $(T_i^\infty)$ and no $(T_i^0)$ miRNA expression] of the top $x$ targets as a function of $x$, we indeed found that the highest scoring targets undergo the largest down-regulation (Fig 4F, dotted lines), as expected. Similar patterns of stronger down-regulation of top targets were also apparent when we sorted targets based on their sensitivity to the miRNA reflected in the $A_F^C$ parameter (Fig 4F, dashed lines). However, the best indicator of the degree of down-regulation of a predicted target was its inferred $K_M$ (Fig 4F, full lines). This could indicate that the inferred $K_M$ values are dominated by $k_{cat}$, the rate of target degradation when complexed to the miRNA, while the rates of miRNA–target association and dissociation vary less

between targets. Finally, we examined what features of the predicted miRNA binding site were most informative for the $A_F^C$, $K_M$ and fold change of the target (Appendix Fig S5B). For this, we selected only the 231 targets with a single binding site (for either of the miRNAs), to ensure that the site context effects could be attributed unambiguously. Consistent with prediction models being trained to predict mRNA level changes upon miRNA transfection, the prediction scores correlate best (in absolute value) with the fold change of the predicted targets in cells with high miRNA expression compared with low miRNA expression. Measures related to the A/U content in the vicinity of sites and their relative location in 3′ UTR are most predictive for $A_F^C$ and $K_M$, whereas the degree of evolutionary conservation is most informative for the fold change of the target.

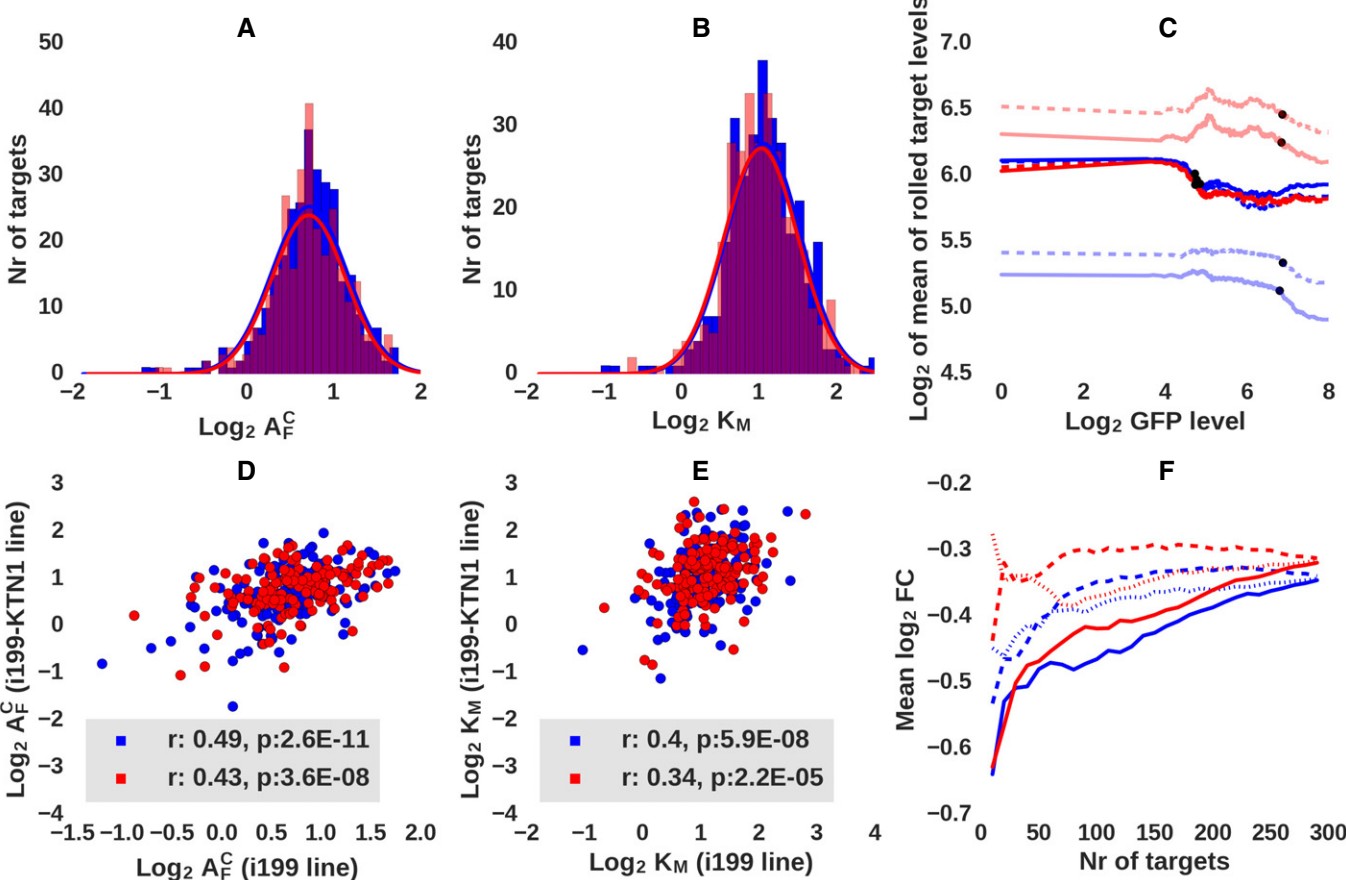

**Figure 4.  Parameters describing the response of individual targets to changes in miRNA expression.**

A, B   Histograms of $A_F^C$ (A) and $K_M$ (B) log$_2$-values of hsa-miR-199a-5p (red) and hsa-miR-199a-3p (blue) targets, inferred from the i199 cell line. The lines indicate the best-fitting Gaussian distributions.

C   Response of hsa-miR-199a-5p (red) and hsa-miR-199a-3p (blue) targets to the miRNAs in i199 cells. Targets were selected based on $A_F^C$ (dashed lines) or $K_M$ (full lines) values, targets with low values of the respective parameters are shown in strong color, and those with high values are shown in faded colors. Twenty targets were summed up for each category. Dots show the point where the targets have undergone ½ of their maximal down-regulation.

D   Scatter plot of log$_2$ $A_F^C$ values inferred for individual targets from the i199 and i199-KTN1 cell lines. Shown are also Pearson's correlation coefficients and corresponding $P$-values.

E   Scatter plot of log$_2$ $K_M$ values inferred for individual targets from the i199 and i199-KTN1 cell lines. Shown are also Pearson's correlation coefficients and corresponding $P$-values.

F   Average log$_2$ fold change of hsa-miR-199a-5p (red) and hsa-miR-199a-3p (blue) targets as a function of the number of top targets considered, where predictions are made based on either $K_M$ values (highest to lowest, full lines), $A_F^C$ values (lowest to highest, dashed lines) or MIRZA-G-C scores (highest to lowest, dotted lines).

## Implications for the ceRNA function of miRNA targets

To evaluate the implications of our results for the debate about the prevalence of competing endogenous RNAs (Denzler *et al*, 2014; Bosson *et al*, 2014), we used again our computational model with realistic $K_M$ values and explored the effect of one miRNA target (the ceRNA) on the expression of all other targets. Target parameters were set as described in section "*In silico* analysis", to cover the range inferred from various experimental systems. We note that a ceRNA is only one species of RNAs expressed in a cell and, for the vast majority of parameter values that are in the range determined for other RNAs in the cell, the ceRNA is predicted to cause expression changes that are very low, below 1%. Nevertheless, we illustrate some of the more interesting scenarios below. We set the decay rate of the free ceRNA to 0.1/h, its $k_{on}$ = 0.2/h, similar to that of other targets, and we varied the $k_{off}$ and $k_{cat}$ to achieve either low or high $K_M$. We then asked how much the expression of the pool of targets with either low (less than 0.02 M) or high (greater than 2 M) $K_M$ targets changes, when the ceRNA is expressed at different levels.

As shown in Fig 5, we found that highly expressed ceRNAs with low $K_M$ can induce the up-regulation of low and especially high $K_M$ targets. However, substantial up-regulation of other targets, larger than a few percent, is only achievable when the ceRNA has very high transcription rate and does not decay when in complex with the miRNA. This is what one intuitively expects, namely that a ceRNA can influence the expression of other targets when its expression is comparable to that of all other targets taken together. On the other hand, if the ceRNA has high $K_M$, its influence on the expression of other targets will be negligible. These results strongly suggest that ceRNAs that were observed so far are highly expressed transcripts that are relatively resistant to degradation. These would be able to "sponge" miRNAs from targets which the miRNA strongly destabilizes, which have high $k_{cat}$ and high $K_M$. Good candidates seem to be the relatively recently described circular RNAs (Memczak *et al*, 2013; Hansen *et al*, 2013). However, given the multiple constraints that a transcript has to fulfill to be able to function as a ceRNA (very high transcription and/or stability, low $K_M$), this mode of regulation should be rare.

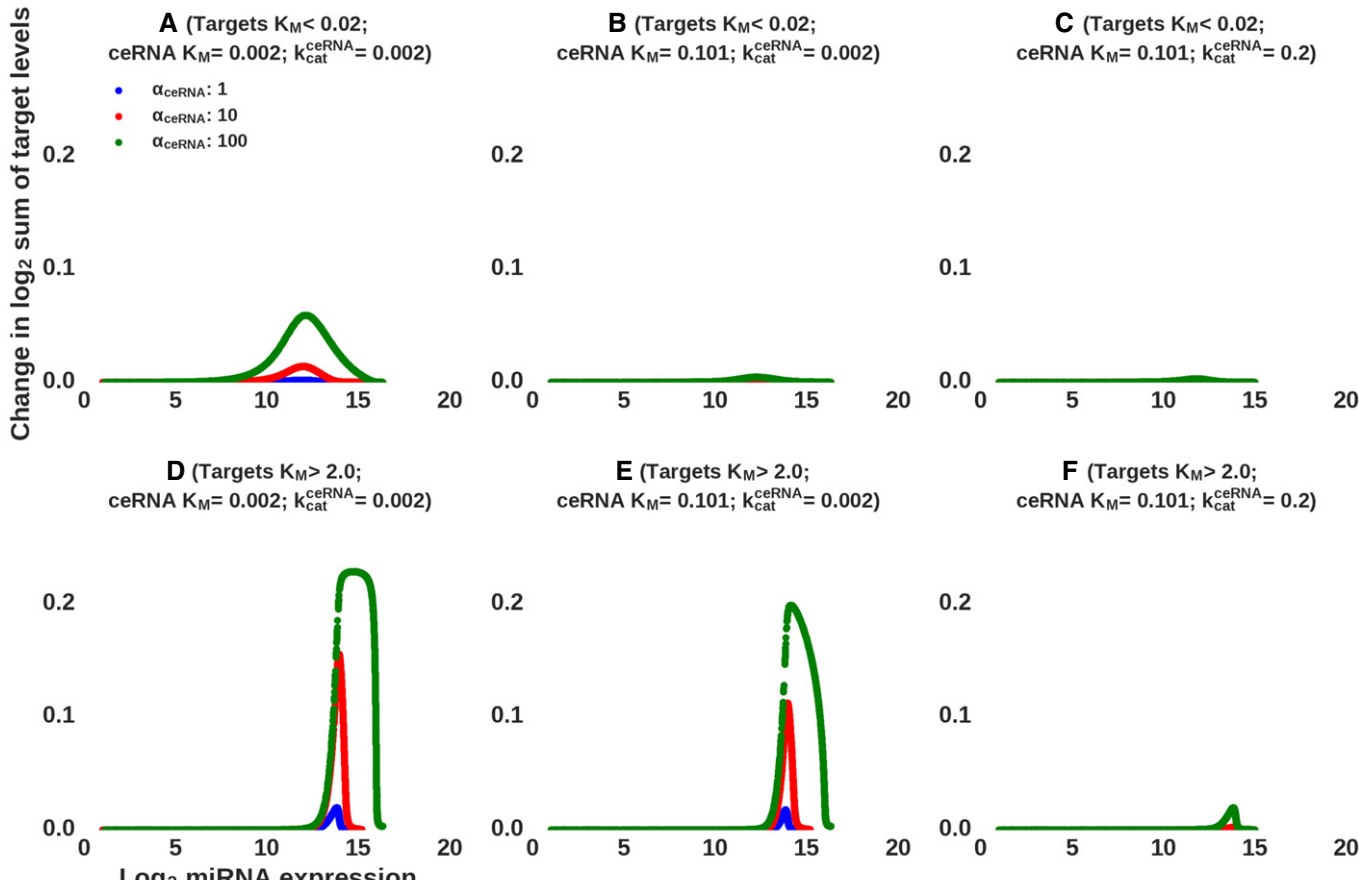

**Figure 5. Predicted response of different types of miRNA targets to the induction of a ceRNA.**

A–F  A competing RNA with low (A, D) or medium (B, C, E, F) $K_M$ is transcriptionally induced at three different levels. (A–C) shows the response of targets with low $K_M$ (< 0.02) to the transcriptional induction of the ceRNA, whereas (D–F) shows the response of targets that have high $K_M$ (> 2.0). The decay rate of the ceRNA when unbound to the miRNA $\delta_{ceRNA}$ was set to 0.1/h, whereas when bound to the miRNA, the ceRNA was assumed to be either stabilized and long-lived ($k_{cat}$ was set to 0.002/h) (A, B, D, E) or destabilized and shorter-lived ($k_{cat}$ was set to 0.2/h) (C, F).

# Discussion

Single-cell RNA sequencing has opened a new route to the quantitative understanding of cell behaviors. This technology has been used to characterize transcript isoforms and gene expression (Shalek *et al*, 2013), to improve classification of cell types (Buettner *et al*, 2015) and to discover new, particularly rare types of cells (Grün *et al*, 2015). The relatively low rate of mRNA capture and the large technical noise remain important issues for single-cell sequencing, particularly with droplet-based methods, which rarely use spike-ins for normalization (Ziegenhain *et al*, 2018; Gao, 2018). However, developments such as unique molecular identifiers (Grün *et al*, 2014) push the boundary toward ever increasing accuracy. Although data analysis methods are still in flux, in our study, we used known properties of miRNA targets to gauge whether our processing of the data is appropriate. For example, we showed that miRNA target down-regulation computed from the inferred target levels in single-cells is similar to the down-regulation inferred from bulk sequencing. Single-cell analysis has also been used to infer parameters of gene expression (see Munsky *et al*, 2015 for a recent review).

Although it was proposed that miRNAs buffer stochastic fluctuations in gene expression between cells (Hornstein & Shomron, 2006), experimental data pertaining to expression of miRNA targets in individual cells with different levels of miRNA expression are very limited. Some studies estimated the effect of endogenous miRNAs on the protein expression noise of target reporters with multiple miRNA-complementary sites (Mukherji *et al*, 2011; Schmiedel *et al*, 2015). The reduction in protein expression noise has been related to the degree of miRNA-induced down-regulation, which is generally limited, except for reporters that carry multiple perfectly complementary miRNA binding sites in their 3′ UTR. Additional studies are needed to evaluate the extent to which miRNAs regulate the expression noise of their targets in their native context (see also Schmiedel *et al*, 2015). Target reporters have also been used to investigate whether miRNAs induce correlations in the expression levels of their targets (Bosia *et al*, 2017). However, how endogenous miRNA targets simultaneously respond to miRNA induction in individual cells is insufficiently understood, leading to ongoing debates about the influence that one target can have on the expression of others.

In this study, we developed a methodology to characterize the regulatory effects of a miRNA on its hundreds of targets in single-cells. We constructed an experimental system in which the expression of a miRNA precursor can be induced with doxycycline together with that of GFP from a bidirectional promoter. This system was initially tested with two protein-coding genes, one for the nerve growth factor and the other for eGFP, which showed good, though not perfect correlation at single-cell level (Bornkamm *et al*, 2005). In our case, absolute quantification of the miRNA and GFP mRNA in cell populations indicated that expression of the two RNAs was highly correlated in response to doxycycline induction, and we thus used the GFP mRNA as a proxy for the miRNA. It is likely that a direct measurement of miRNA expression in the cells whose mRNAs are sequenced would further increase the accuracy of the results of our model, and we expect the technology to become available in the near future. We showed that this system exhibits predicted behaviors such as a peak in target noise as well as a peak in the correlation of target levels, in the region of maximal sensitivity to the miRNA. The construct can be easily modified to enable inducible expression of other miRNAs. We further developed a methodology for the variational fitting of Michaelis–Menten-type constants ($K_M$) characterizing individual miRNA targets. This method takes advantage of the variability in transcriptional activity that leads to variability in miRNA expression levels between cells. For the first time, we have uncovered the hierarchy of targets of a miRNA, defined by the miRNA concentrations at which these targets respond within the context of all other targets in the cell as well as by the Michaelis–Menten-type constants. We found that high $K_M$ targets undergo the largest down-regulation, indicating that this parameter reflects primarily their $k_{cat}$, the rate of decay in the presence of the miRNA. Some targets were particularly sensitive to the miRNA, requiring relatively low miRNA concentrations to respond and having reproducibly low $A_F^C$ values. Their higher prediction scores and enrichment in DNA-binding factors suggest that these are prototypical miRNA targets (Gruber & Zavolan, 2013). Simulations indicate that targets with low $K_M$ and low $A_F^C$ values could sequester the miRNA from other targets if they are highly expressed and do not decay substantially when they interact with the miRNAs. Current approaches for studying miRNA–target interactions that measure mRNA level changes upon miRNA overexpression to uncover the most relevant targets likely overlook these targets. Thus, it would be interesting to apply our approach to systems in which functional ceRNAs have been reported (Poliseno *et al*, 2010; Cesana *et al*, 2011; Cheng & Lin, 2013). Interestingly, early analyses of miRNA and target expression found that many miRNA targets are expressed at relatively high level in the tissue in which the miRNA is expressed (Farh *et al*, 2005). However, this has been attributed to miRNAs optimizing the protein output of their targets rather than entirely suppressing it. Our analysis also suggests that targets with low $A_F^C$, which bind the miRNAs but do not undergo substantial down-regulation in response to it, could impose a threshold for miRNA-dependent regulation, which would otherwise affect a large fraction of the transcriptome.

To demonstrate the robustness of our approach, we have inferred parameters of individual targets from two closely related cell lines. However, it is likely that the sensitivity of a target to miRNA regulation is context-dependent (Erhard *et al*, 2014). Because we wanted to map the parameters of miRNA–target interaction in a native context of mRNA expression, we induced the miRNA expression from an exogenous construct in HEK 293 cells. Although a large number of studies of miRNA-dependent gene regulation have similar designs, it remains possible that the "true" targets of the miRNA are not naturally expressed in the cell type in which the experiment is carried out. To fully address this possibility, one would perhaps have to progressively remove a highly abundant, cell type-specific miRNA, which would be more challenging than inducing miRNA expression. miR-122 in liver cells could be a good candidate for this type of experiment (see also Denzler *et al*, 2014).

The miRNA target parameters that we inferred in our study will enable an improved understanding of the dynamics of networks containing many competing miRNA targets. Furthermore, the approach can be easily extended to RNA-binding protein regulators of mRNA stability as well as to other types of regulators such as transcription factors.

# Materials and Methods

## A model to describe the dynamics of miRNA targets

We used the model from equation (1) in the main text and also shown below, which considers $M$ targets of a miRNA, each being described by a transcription rate $\alpha_i$, decay rate $\delta_i$, rate of binding the Ago-complexed miRNA $k_{on_i}$, rate of dissociating from this complex $k_{off_i}$ and rate of degradation when in the complex $k_{cat_i}$. With $m_i$ being the concentration of the free target, $A_{m_i}$ the concentration of the miRNA-bound target, and $A$ the total concentration of Ago-miRNA complexes, we have the following system of $2M$ differential equations

$$\frac{\partial m_i(t)}{\partial t} = \alpha_i - \delta_i m_i(t) - k_{on_i} m_i(t)\left(A - \sum_{j=1}^{M} A_{m_j}(t)\right) + k_{off_i} A_{m_i}(t)$$
$$\frac{\partial A_{m_i}(t)}{\partial t} = k_{on_i} m_i(t)\left(A - \sum_{j=1}^{M} A_{m_j}(t)\right) - k_{off_i} A_{m_i}(t) - k_{cat_i} A_{m_i}(t).$$
$$(1)$$

Denoting the total concentration of mRNA $i$ (either free or bound to the miRNA) by $T_i$ and summing the two equations corresponding to mRNA $i$, the dynamics of $T_i$ is described by

$$\frac{\partial T_i(t)}{\partial t} = \alpha_i - \delta_i m_i(t) - k_{cat_i} A_{m_i}(t), \qquad (2)$$

or, in terms of the fraction $f_i$ of molecules of mRNA $i$ that are bound to miRNAs,

$$\frac{\partial T_i(t)}{\partial t} = \alpha_i - \delta_i (1 - f_i) T_i(t) - k_{cat_i} f_i T_i(t). \qquad (3)$$

Defining the total concentration of mRNA $i$ when no miRNA is present as $T_i^0 = \frac{\alpha_i}{\delta_i}$ and when the miRNA is in high excess as $T_i^\infty = \frac{\alpha_i}{k_{cat_i}}$, we obtain the total concentration of mRNA $i$ at a steady state as

$$T_i^* = \frac{\alpha_i}{\delta_i(1 - f_i) + k_{cat_i} f_i} = \frac{T_i^0}{1 + f_i\left(\frac{T_i^0}{T_i^\infty} - 1\right)}. \qquad (4)$$

Note that the concentration of the miRNA is reflected in the fraction of bound targets. In our experimental system, we vary the expression of the miRNA from very low to very high levels and we can therefore estimate $T_i^0$ and $T_i^\infty$. However, the fraction of mRNA $i$ that is bound to the miRNA depends not only on the constants of interaction of this mRNA with Ago-miRNA complexes, but also on all other targets that are present in the system. To determine the interaction constants, we first derive for each mRNA $i$ the fraction $f_i$ that is bound to the miRNA, as follows. At equilibrium, we have

$$m_i k_{on_i} A_F = A_{m_i}\left(k_{off_i} + k_{cat_i}\right), \qquad (5)$$

$$\frac{m_i}{A_{m_i}} = \frac{1 - f_i}{f_i} = \frac{k_{off_i} + k_{cat_i}}{k_{on_i} A_F}, \qquad (6)$$

and thus

$$f_i = \frac{1}{1 + \frac{K_{M_i}}{A_F}}, \qquad (7)$$

with the Michaelis–Menten parameter defined as $K_{M_i} = \frac{k_{off_i} + k_{cat_i}}{k_{on_i}}$. Considering all cells $j \in \{1, \ldots, N\}$, each with a different concentration of free Ago-miRNA complexes $A_{F_j}$, and substituting $f_i$ in equation (4), we obtain

$$T_{ji} = \frac{T_i^0}{1 + \frac{1}{1 + \frac{K_{M_i}}{A_{F_j}}}\left(\frac{T_i^0}{T_i^\infty} - 1\right)}, \qquad (8)$$

where $T_{ji}$ is the total concentration of mRNA $i$ in cell $j$, which can be computed from the measured target levels. We isolate the ratio $\frac{K_{M_i}}{A_{F_j}}$ and rewrite

$$\frac{\frac{T_i^0}{T_i^\infty} - 1}{\frac{T_i^0}{T_{ji}} - 1} - 1 = \frac{K_{M_i}}{A_{F_j}} \qquad (9)$$

or equivalently, in vector form, substituting the left-hand side of the equation by $\tilde{T}$,

$$\tilde{T} = A_F^{-1} K_M. \qquad (10)$$

Here, $K_M$ is a $(1 \times M)$ matrix of Michaelis–Menten constants for individual mRNAs, $A_F^{-1}$ is a $(N \times 1)$ matrix of free Ago-miRNA complexes in individual cells, and $\tilde{T}$ is a $(N \times M)$ matrix of expression levels of individual mRNAs in individual cells. $\tilde{T}$ can be viewed as a Kronecker product of the two vectors $K_M$ and $A_F^{-1}$ written in a more general form as

$$B = xy^\top. \qquad (11)$$

Determining the vectors $x$ and $y$ becomes the reverse Kronecker product problem and has a known solution satisfying

$$\min_{x,y} \|B - xy^\top\|_F, \qquad (12)$$

where $\|\cdot\|_F$ denotes the Frobenius norm. The solution is obtained from the singular value decomposition (SVD) $B = U\Sigma V^\top$ as

$$x_i = \sqrt{\Sigma_{11}} U_{i1}, \quad y_i = \sqrt{\Sigma_{11}} V_{i1}. \qquad (13)$$

From equation (9), we see that the SVD provides us the solution $(A_F, K_M)$, up to a scaling factor $a$, $\frac{aK_M}{aA_F} = \frac{K_M}{A_F} \forall a \in \Re$. In principle, it is possible to determine the factor $a$, which explains the data best, using the total concentration of Ago-miRNA complexes $A$ in all cells.

Fitting the vectors $A_F$ and $K_M$ on data generated from simulations of model (1), we found that the correlation of the fitted $A_F$ with the input value was significantly higher than for $K_M$. This is explained by the fact that we use the total concentration of the miRNA in the cells to sort the cells and smoothen the mRNA expression. $A_F$ being a monotonic, strictly increasing, continuous function of $A$, smoothing the data along the cell dimension (i.e., along the $j$ index in equation (8)) leads to a reduction in noise in the direction of the miRNA levels $A_F$ but not in the mRNA dimension $K_M$. Therefore, the vector $A_F$ is inferred more precisely compared to $K_M$. Using the more precisely inferred $A_F$ values and averaging over cells, we can

increase the precision of $K_{M_i}$ values; relation (10) always holds, and after fitting, we use the values of $A_F$ to compute the values $K_{M_i}$ by averaging $A_{F_j} \tilde{T}_{ji}$ over all cells $j = 1 \ldots N$

$$K_{M_i} = \frac{1}{N} \sum_{j=1}^{N} A_{F_j} \tilde{T}_{ji}, \ j = 1, \ldots N. \quad (14)$$

We define $A_{F_j}^C$ the concentration of free Ago at which the target will be exactly halfway between $T_i^0$ and $T_i^\infty$.

$$\frac{T_i^0}{1 + \frac{1}{1 + \frac{K_{M_i}}{A_{F_j}^C}} \left( \frac{T_i^0}{T_i^\infty} - 1 \right)} = \frac{T_i^0 + T_i^\infty}{2} \Rightarrow A_{F_j}^C = \frac{K_{M_i}}{\frac{T_i^0}{T_i^\infty}}.$$

### *In silico* analysis

Stochastic simulations based on equation (1) were used to verify the solution obtained in equation (8). Stochastic simulations were performed using StochKit v.2.0.11 (Sanft *et al*, 2011) with a tau-leaping algorithm. For each *in silico* cell, six simulations of length 100,000 (arbitrary time units) were carried out to ensure convergence. The first 10,000 steps were considered the "burning phase" and were discarded before the analysis. Means and standard deviations were calculated from the values obtained in the independent simulations.

To test the $K_M$ inference method, we constructed an *in silico* data set as follows. We considered a regulatory network of 300 miRNA targets. Each target was characterized by parameters $\alpha_i$, $\delta_i$, $k_{cat_i}$, $k_{on_i}$, $k_{off_i}$, whose values were assumed to be in the ranges provided by our previous literature survey (Hausser & Zavolan, 2014). For each target, we chose a set of parameters from log-normal distributions, which are shown in Appendix Fig S4. Similar to the experimental data set, we considered 4,000 virtual cells, each with a distinct concentration of free Ago-miRNA complexes, chosen from a uniform distribution on the $\log_2$ range of −40 to 14, such as ~50% of cells end up with no miRNA expression, as observed in the experiment. The expression of all targets as a function of the miRNA abundance in these virtual cells is presented in Fig 3A. Note however that in the experimental system, we could not measure miRNA levels but rather the copy number of the GFP mRNA, and thus, in comparing the response of targets in the *in silico* and experimental systems, the *x*-axes differ, being the miRNA level for the *in silico* data and the GFP mRNA level for the experimental data. Interestingly, the miRNA-to-GFP mRNA conversion factor corresponds well with the miRNA:GFP mRNA ratio of 4–8 that is apparent from the qPCR data (see also Fig 1). Each target starts to decay at a specific threshold, depending on its parameters of interaction with the miRNA and the effective miRNA concentration, which depends on the other targets as well. To complete our *in silico* data generation, we added log-normal noise to the analytically computed expression levels of the targets (see Fig 3C).

To focus on cells where the miRNA targets responded most sensitively to the miRNA, we started with the selection of single-cells from which to construct the matrix $\tilde{T}$. $T_i^\infty$ and $T_i^0$ were calculated from about 200 *in silico* cells with the highest and 1,600 cells with the lowest concentration of miRNA, numbers similar to these in the experimental system. We analyzed the derivative of the sum of $\log_2$

target levels in function of miRNA expression and selected the cells where the gradient was lower than −0.01 (Fig 3B). Cells with target expression values very close to $T_i^0$ or $T_i^\infty$ were filtered out to avoid instabilities caused by division by small numbers (see equation (9)). Next, we applied a smoothing procedure to ensure that at intermediate miRNA expression, the $T_{ji}$ level of targets $i$ in cells $j$ was strictly in the range $(T_i^\infty; T_i^0;$ see Fig 3C). We started by replacing the expression level of a given target in a given cell with the mean over the 50 cells with miRNA expression level closest to that in the reference cell. In a second pass, for the smoothed $T_{ij}$ values outside of the $(T_i^\infty; T_i^0)$ range, we computed again a running mean starting with a window size of ten and discarding iteratively the strongest outliers until the mean value $T_{ij}$ within each window was within the $(T_i^\infty; T_i^0)$ range. For the windows where this procedure did not leave any points, we increased the size of the second-pass window locally, repeating the pruning procedure until all the $T_{ij}$ values were within the $(T_i^\infty; T_i^0)$ range. To ensure the stability of the SVD, we adjusted the boundary of the $T$ intervals computed from the data by a small safety margin $c$ (i.e., $T_i^0 - c > T_{ij} > T_i^\infty + c$, $c = 10\%$ of $T_i^0 - T_i^\infty$ for each gene).

We assessed the accuracy of the fitting procedure by comparing the inferred $A_{F_j}^C$ and $K_{M_i}$ parameters with those that were used in the model that generated the *in silico* data. In spite of very high noise (Fig 3C), there was a good correlation between the fitted and input values of the parameters, as shown in Fig 3D. In addition, the correlation of parameters observed when simulating two independent "samples", with two independent noise applications, was also relatively high (Fig 3D). We also observed that the range of inferred $K_M$s is narrower than the range of input $K_M$s.

Having validated our inference procedure on *in silico* data, we applied it to the experimental data.

### Cell culture

We used a human epithelial kidney (HEK) 293 cell line with inducible expression of hsa-miR-199a (i199) and a derivative of this cell line (i199-KTN1) in which a Renilla luciferase coding sequence followed by the 3′ UTR of the kinectin 1 gene (KTN1) was inserted in the genome. These cell lines have been introduced in a previous study (Hausser *et al*, 2013). Cells were grown in DMEM with 10% FCS supplemented with Pen-Strep and Hygromycin for plasmid integrity. For all the experiments, unless otherwise mentioned, cells were stimulated with doxycycline at concentrations of 1, 0.3, 0.1, 0.03, 0.01, 0.003, 0.001, 0.0003, or 0 μg/ml, for 8 consecutive days. During this period, fresh medium with doxycycline was provided every 24 h and cells were split every 72 h to prevent the slow-down growth in confluent cultures (Ghosh *et al*, 2015).

### Single-cell mRNA sequencing

#### *Cell capture, GEM Barcoding, and cDNA synthesis*
Cells were detached with Accutase® Reagent (Gibco, Life Technologies™). The cell number was determined with the Countess™ Automated Cell Counter (Invitrogen™) following manufacturer's instructions. Cells that were induced with different doxycycline conditions (see section above) were pooled together in equal proportions (1,500 cells/μl of each). The cells were finally resuspended in PBS containing 0.04% BSA at a target concentration of

700 cells/µl after straining with a cell strainer to avoid clumps. This is performed so as to partition the input cells across tens of thousands of droplets (GEMs) for the purpose of lysis and barcoding. GEM Generation and Barcoding was performed according to manufacturer's instructions (Chromium™ Single-Cell 3′ Reagent Kits v2, Part No-120234, 10x Genomics). Subsequently, reverse transcription (RT) and post-GEM-RT cleanup were done exactly as per manufacturer's protocol. The purified GEM-RT product was then pre-amplified for 10 cycles, purified with SPRI select (Beckman Coulter), and analyzed on a High Sensitivity Bioanalyzer.

### Library preparation and sequencing

Library construction including Fragmentation, End Repair, and A-tailing was performed as per manufacturer's protocol (Chromium™ Single-Cell 3′ Reagent Kits v2, Part No-120234, 10x Genomics). Subsequently, the fragments were purified with a double-sided size selection with SPRI select (Beckman Coulter) and ligated to adapters. After ligation, the samples were purified once more with SPRI select prior to the steps of sample index PCRs. The end product was finally obtained with another round of double-sided SPRI selection of the PCR. Quality control of the libraries was done on an Agilent Bioanalyzer High Sensitivity Chip. Libraries were then sequenced (Paired End) on a NextSeq 500 system [NextSeq 500/550 High Output v2 Kit (75 cycles)], and the reads were obtained according to the following parameters:

(i)   Seq Read 1, 26 cycles;(ii)   Seq Read 2, 58 cycles;
(iii)   IDX Read, 8 cycles;
(iv)   Illumina base-calling software version: bcl2fastq v2.19.0.316; and
(v)   Demultiplexing software version: cellranger mkfastq (2.0.0).

The library preparation and sequencing were performed at the Genomics Facility Basel. The sequencing data have been deposited to the Sequence Read Archive (www.ncbi.nlm.nih.gov/sra/) under the accession number SRP067502.

### Computation of the coefficient of variation of target expression

Given the set of cells sorted by their GFP expression, we calculated the coefficient of variation ($C_V$, standard deviation/mean) of a specific target as follows. We traversed the list of cells from those with lowest to those with highest GFP expression, and for each cell, we considered the 199 cells with closest GFP level to the reference cell and calculated the $C_V$ of each target. We then log₂-transformed the $C_V$ of individual targets and determined the mean (and standard error) over all 100 selected low $A_F^C$ targets. We applied the same procedure to all non-targets (genes targeted neither by hsa-miR-199-3p nor by hsa-miR-199-5p). We then subtracted the log₂ mean $C_V$ of targets and non-targets, repeated this procedure for the entire GFP expression range and show the normalized $C_V$ as a function of the log₂ GFP level in the reference cell.

### PAGODA variance normalization

The i199 and i199-KTN1 single-cell data sets were divided into 100 cell batches, grouped according to GFP expression level in the cells. A random sample of 100 cells was subsampled from the cell population with no GFP expression. Next, the PAGODA data preparation, error modeling, and variance normalization functions were used with standard parameters, on the raw data sets, as specified in the PAGODA tutorial, http://hms-dbmi.github.io/scde/pagoda.html.

The normalized variance used for the analysis is shown in Appendix Fig S3B and E.

### Computation of the pairwise correlation coefficients of target expression levels

Given a population of cells sorted by their GFP expression, we calculated the Pearson correlation of log₂ expression levels for all pairs of 100 targets, in function of GFP level (as for $C_V$, average values were computed over 199 cells with GFP expression closest to that in the reference cell). Thus, we started from those cells with lowest GFP expression and moved by one cell at a time to cells with the highest GFP expression, computing the mean correlation coefficient (and standard error of the mean) over all pairs of genes within a cell. We repeated the procedure for 50 evaluations of 100 random genes that were not predicted as targets. Finally, we divided the mean correlation coefficients of targets and non-targets and shown this as function of GFP level in the cell.

### Computation of GO enrichment

The hyperGTest function from GOstats package (R Bioconductor repository) was used to find enriched GO terms. The maximum "*PvalueCutoff*" for reporting was set to 0.1, "conditional" to "*TRUE*", and "testDirection" to "Over".

## Cell population mRNA-Seq

### Total RNA isolation

Total RNA was extracted with TRI Reagent® (Sigma-Aldrich) following manufacturer's instructions. Briefly, cells were detached from the plate by 5-min incubation with Trypsin–EDTA solution (T3924; Sigma-Aldrich), conditioned media were added, and whenever necessary, cells were counted with a Countess™ Cell Counter (Thermo Fisher Scientific). A defined number of cells were pelleted and either snap-frozen for future use or resuspended right away in TRI Reagent ® (#T9424; Sigma-Aldrich). Total RNA was resuspended in nuclease-free water (#AM9937; Thermo Fisher Scientific). Samples were always kept on ice or at −80°C.

### mRNA purification

To select the Poly(A)⁺ RNA, a double purification with Dynabeads® Oligo (dT)25 (Dynabeads® mRNA DIRECT™ Kit, Ambion™) was performed, using the manufacturer's manual and recommendations. Since the starting material was purified total RNA, only buffer B was used for the washing steps.

### Library preparation

Purified mRNA was fractionated with alkaline hydrolysis buffer at 95°C for 5 min. Fractionated mRNA was selected with RNeasy MinElute Cleanup Kit (Qiagen, Inc.). Purified mRNA fragments were dephosphorylated with FastAP (Life Technologies, Inc.) and 5′-phosphorylated with PNK (Life Technologies, Inc.) following manufacturer's instructions for optimal conditions of the enzymatic reaction. After another round of RNeasy MinElute Cleanup Kit (Qiagen, Inc.), a pre-adenylated DNA adapter (5′-TGGAATTCTCGGGTGCCAAGG-3′) was ligated to the 3′ end of the mRNA fragments at 4°C overnight using the T4 RNA ligase 2, truncated K227Q (New England Biolabs, Inc.), in 1× T4 RNA ligase buffer (no ATP) and 15% DMSO. The next day, after another round of RNeasy MinElute

Cleanup Kit (Qiagen, Inc.), an RNA adapter (5′-GUUCAGAGUUCUA CAGUCCGACGAUC-3′) was ligated to the 5′ end of the RNA fragments at 4°C overnight using the T4 RNA ligase 1 (Life Technologies, Inc.), in 1× T4 RNA ligase buffer (1 mM ATP) and 15% DMSO. Next day, after another round of RNeasy MinElute Cleanup Kit (Qiagen, Inc.), reverse transcription was performed using SuperScript IV (Invitrogen, Inc.) and RTP primer (5′-CCTTGGCACCCGAGAATT CCA-3′) following manufacturer's instructions. cDNA was then amplified by 12 cycles of PCR using NEBNext® High-Fidelity 2× PCR Master Mix (NEB, Inc.) and Illumina TruSeq® Small RNA PCR compatible primers.

*Library sequencing*
The library was sequenced in the Genomics Facility Basel, on Illumina HiSeq 2000 or HiSeq 2500 instruments using TruSeq compatible primers. Reads of 50 nt were generated along with 8 nt index reads corresponding to the sample-specific barcode.

## Read mapping and data preprocessing

Reads from single-cell and cell population mRNA-Seq were mapped to the transcriptome (Ensembl, GRCh38.rel88) with Cellranger-1.3.1, the software provided by 10x Genomics to map the reads produced by the Chromium™ Single-Cell 3′ solution. Cellranger processes the cell and transcript barcodes, uses STAR 2.5.1b to align the reads, and counts the number of transcripts observed from each gene to provide a table of unique molecular identifier (UMI) counts per gene and per cell. The sequence of the eGFP mRNA that was expressed from the exogenous pRTS1 vector was added to the transcriptome before mapping. After summing the counts for all Ensembl entries for a given Entrez gene ID, the gene counts were normalized to have in each cell one million counts. Next, a pseudo-count, 0.001, was added to each gene (and 1.0 to GFP gene for clarity of visualization). In all of the analyses, genes with very low final estimated expression (mean TPM < 7 across cells) were discarded.

*Targets selection*
If not specified otherwise, we used in analyses the 300 highest probability targets predicted by MIRZA-G-C (Gumienny & Zavolan, 2015) that were down-regulated at least 8% at the maximum miRNA concentration $[\log_2 (T_i^\infty / T_i^0) < -0.12]$. This selection applied to both miRNAs and both cell lines.

## mRNA and miRNA qPCR

Cells were induced with various concentrations of doxycycline (as indicated in the figure) for 8 days. After counting the cells, total RNA was extracted with TRI Reagent® (Sigma-Aldrich) following manufacturer's instructions. cDNA of the targets of interest was generated using SuperScript III (Invitrogen™) following manufacturer's protocol. For miRNA assays, reverse transcription and PCR of either non-induced or Dox-induced cells were performed following the TaqMan® Small RNA Assays quick reference protocol (Life Technologies™) with 100 ng of total RNA. For estimation of relative miRNA quantities, hsa-miR-16 levels were used as an invariant control. For reverse transcription of GFP mRNA, the following linear DNA primer was used: EGFP_R RT TaqMan, 5′-TGTCGCCCTC GAACTTCAC-3′. To generate a cDNA copy of hsa-miR-199a-5p,

a stem-loop primer system from Life Technologies™ was used (Assay ID-000498). All qPCRs were performed and read in StepOnePlus™ Real-Time PCR Systems (Life Technologies™). To obtain absolute quantification data, standard curves for GFP and hsa-miR-199a-5p were also included. GFP mRNA was generated by *in vitro* transcription with pcDNA3-eGFP linearized vector and RiboMAX™ Large Scale RNA Production System—T7 (Promega, Co.) following manufacturer's instructions. Molarity was estimated taking into account mass concentration (Qubit® RNA HS Assay Kit, Life Technologies™), average length (Agilent RNA 6000 Pico Kit, Agilent Technologies, Inc), and fragment sequence, with the following formula: molarity = mass/(length × mass RNA base). The hsa-miR-199a-5p miRNA (5′-CCCAGUGUUCAGACUACCUGUUC-3′) was ordered from Microsynth AG, and the molarity was calculated the same way. Absolute molecule numbers were obtained utilizing the StepOne™ Software (Life Technologies™).

## CLIP-Seq

CLIP-Seq was performed as described in Jaskiewicz *et al* (2012) with few modifications. Ago2-CLIP in i199 cells was performed using Ago2 antibody-containing serum (kind gift from Prof. Gunter Meister, University of Regensburg, Germany) crosslinked to 100 μl of Dynabeads Protein G (#10003D; Thermo Fisher Scientific). TURBO DNase (#AM2238; Thermo Fisher Scientific) treatment of UV-crosslinked cell lysates was followed by a brief treatment with RNase T1 (#EN0541; Thermo Fisher Scientific) for the specific recovery of Ago2-protected RNA fragments. Subsequently, antibody-bound beads were incubated with the cell lysate for 3 h at 4°C for precipitation. Furthermore, the beads were washed, treated again with RNase T1, dephosphorylated, and labeled with radioactive ATP [γ-32P] to facilitate purification of the required fragments from a nitrocellulose membrane filter following a standard SDS–PAGE electroelution process. The recovered RNA fragments were ligated to a pre-adenylated 3′ adapter, annealed to the RT primer, and subsequently ligated to the 5′ adapter prior to a step of reverse transcription with SuperScript™ III Reverse Transcriptase (#18080044; Thermo Fisher Scientific). In the finals steps, a PCR amplification of the reverse-transcribed cDNA derived from the Ago2 immunoprecipitate was followed by size selection of 140–180 nucleotide long fragments in native PAGE and sequenced after purification.

# Data availability

The data sets produced in this study are available in the following databases:

- RNA-Seq, scRNA-Seq, and CLIP-Seq data: NCBI Sequence Read Archive with accession SRP150046 (https://www.ncbi.nlm.nih.gov/Traces/study/?acc=SRP150046).

**Expanded View** for this article is available online.

## Acknowledgements

We are grateful to Andrea Riba, Alexander Kanitz, Joao Guimaraes, Andreas R. Gruber, and the other members of the Zavolan group for providing input and feedback during the project and for the careful reading of the manuscript.

The work was supported by the European Research Council Starting grant 310510-WHYMIR and by the SystemsX.ch systems biology initiative in Switzerland through the RTD grants 51RT-0_145680 (StoNets) and 51RT-0_145728 (NeuroStemX).

## Author contributions

AJR co-developed the mathematical model and carried out the simulations and computational analysis of the experimental data; SG generated the single-cell sequencing data and carried out experiments to validate miRNA expression; JB developed the mathematical model; AV-V helped set up the experimental system and carried out the CLIP experiments; APS developed the i199 and i199-KTN1 cell lines and helped with the CLIP experiments; AJG analyzed the down-regulation of miRNA targets in bulk populations; KE and CB provided technical help with the single-cell experiments; EvN contributed to the mathematical model; MZ designed the study; contributed to data analysis, and supervised the work; AJR and MZ wrote the manuscript with help from all authors.

## Conflict of interest

The authors declare that they have no conflict of interest.

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
