## [Review Process File · Molecular Systems Biology]

Single cell mRNA profiling reveals the hierarchical response of miRNA targets to miRNA induction

Andrzej J. Rzepiela, Souvik Ghosh, Jeremie Breda, Arnau Vina-Vilaseca, Afzal P. Syed, Andreas J. Gruber, Katja Eschbach, Christian Beisel, Erik van Nimwegen and Mihaela Zavolan.

Review timeline:

Submission date:	6 th February 2018
Editorial Decision:	9 th March 2018
Revision received:	15 th June 2018
Editorial Decision:	18 th July 2018
Revision received:	31 st July 2018
Accepted:	3 rd August 2018

Editor: Maria Polychronidou

Transaction Report:

1st Editorial Decision

9th March 2018

Thank you again for submitting your work to Molecular Systems Biology. We have now heard back from the three referees who agreed to evaluate your study. As you will see below, the reviewers appreciate that the study seems potentially interesting for the miRNA research community. Reviewer #1 is not convinced about the conceptual novelty of the study. However, during our pre-decision cross-commenting process, in which the referees are given the chance to comment on each other's reports, reviewers #2 and #3 emphasized that in their opinion, the novelty of the study lies in the use of scRNA-seq data in combination with the mathematical model of miRNA mediated regulation.

The reviewers raise a series of concerns, which we would ask you to address in a major revision. Without repeating all the points listed below, some of the more fundamental issues raised are the following:

- The effect of biochemical features of miRNA-mRNA interactions should be examined and discussed.
- The effect of targets sharing multiple miRNAs needs to be considered.
- As reviewer #3 mentions, the robustness of the methods used for the normalization of the data should be better demonstrated.
- The work needs to be better placed in the context of previous work and further relevant citations should be included.

Of course all other issues raised by the referees would need to be thoroughly addressed.

REFeree REPORTS

Reviewer #1:

Rzepiela et al. propose a simple M-K kinetic model for miRNA regulation of multiple targets and fit this model to results from biochemical assays generated for this manuscript. Biochemical assays include the upregulation of has-miR199-a in 293 cells and a sister cell type overexpressing the 3'UTR of KTN1, following by scRNA-Seq. Based on their model and experiment, the authors conclude that (1) they have obtained the first response parameters for miRNA targets, and (2) that miRNA correlate target expression and increase the variability of target expression across cells. I find limited novelty in the finding of this manuscript and don't accept the claims for originality, as presented by the authors. In my view, the manuscript's novelty and originality are insufficient for publication at MSB. Major issues are listed below

1. The authors neglected to cite previous work on the topic, including (Arvey et al., 2010; hyun Kim et al., 2013; Luna et al., 2015). In particular, both there has been a slew of proposed kinetic models for miRNA regulation that are similar in nature to what was proposed here (Chen et al. 2016, Vasilescu et al. 2016, Cesana et al. 2013, Yuan et al. 2015, Ala et al. 2013, Bosia et al. 2013, and others...). Some, including Luna et al. and Bosson et al., had experimental data that could be used to fit these models. The conclusions here are mostly in line with their results.
2. While the authors address the issue of multiple targets per miRNA, they do not address issues that arise from targets sharing multiple miRNAs (Chiu et al., 2015). Can the model be extended to account for both multiple targets and regulators? Without this, conclusions about the expected frequency of ceRNA regulation are suspect.
3. The authors are studying cells in culture. Can the authors argue for the benefit of using scRNAs that produce fewer reads per cells in this homogeneous system? I find the idea that perturbations lead to variability across cells to be both natural and unsurprising.
4. On a minor note, the authors elected to study 300 targets predicted by their program. How do these compare to responses by other genes?

References

- Arvey, A., Larsson, E., Sander, C., Leslie, C.S., and Marks, D.S. (2010). Target mRNA abundance dilutes microRNA and siRNA activity. *Molecular systems biology* 6, 363.
- Chiu, H.S., Llobet-Navas, D., Yang, X., Chung, W.J., Ambesi-Impiombato, A., Iyer, A., Kim, H.R., Seviour, E.G., Luo, Z., Sehgal, V., et al. (2015). Cupid: simultaneous reconstruction of microRNA-target and ceRNA networks. *Genome Res* 25, 257-267.
- hyun Kim, D., Grün, D., and van Oudenaarden, A. (2013). Dampening of expression oscillations by synchronous regulation of a microRNA and its target. *Nature genetics* 45, 1337.
- Luna, J.M., Scheel, T.K., Danino, T., Shaw, K.S., Mele, A., Fak, J.J., Nishiuchi, E., Takacs, C.N., Catanese, M.T., de Jong, Y.P., et al. (2015). Hepatitis C Virus RNA Functionally Sequesters miR-122. *Cell* 160, 1099-1110.

Reviewer #2:

In "Single cell mRNA profiling reveals the hierarchical response of miRNA targets to miRNA induction," the authors utilize both experimental and modelling approaches to assess the impact of miRNA on mRNA levels. The primary experimental approach is the introduction of an inducible bidirectional vector expressing both GFP and pri-miRNA (hsa-miR-199a-5p/3p) into HEK 293 cells, with measurement of GFP mRNA then serving as a surrogate for miRNA expression. The authors then assess the HEK 293 cells both by bulk sequencing and single cell sequencing at different levels of GFP (miRNA) expression and assess the impact on predicted miR-199 targets. This experimental approach is complemented by modelling considering the dynamics of free mRNA and Ago/miRNA bound mRNAs in various contexts.

The work attempts to address several aspects of miRNA biology, including the previous observations that only a subset of predicted mRNA targets are observed to change upon miRNA

overexpression and the somewhat conflicting viewpoints surrounding the idea that mRNAs compete for Ago-miRNA binding such that changes in levels of an endogenous gene can influence the totality of bound targets (ceRNA hypothesis). The authors generate a dataset in which miRNA levels vary across a range of expression within a cell population and then use single cell mRNA analysis to determine the impact on targets. To our knowledge this dataset is novel and of interest to the greater miRNA community (and likely the modelling/inference community as well). The authors then apply various modelling techniques to define two parameters K_m and ACF that define the sensitivity of a particular mRNA to miRNA (in this case hsa-199) and argue the range of observed and predicted parameters defines a hierarchy of miRNA-mRNA sites. While the assertion that miRNA sites exist in a hierarchy of occupancy or activity is not new, their inference of this hierarchy from empirical single cell sequencing data represents an interesting approach. Where this work falls a bit short in its current form is in its failure to link the observed and inferred hierarchy here with biochemical features of miRNA-mRNA interactions, such as the idea that binding site type (6- vs 7- vs 8-mer) or other such features (ΔG and/or thermal stability of interaction) might dictate Ago occupancy.

Major points:

1. As mentioned, the hierarchy of sites is not related to any features of miRNA-mRNA interaction beyond the MIRZA-G-C score. This would greatly add to broadening the interest of the findings. Additionally, it is unclear why the authors chose this ranking algorithm as opposed to TargetScan or biochemically defined miRNA-mRNA interactions (for example through cross linking immunoprecipitation experiments, if this data is available for HEK293).
2. Is there endogenous miR-199 in these cells, and if so what are its levels? The authors need to address whether the fundamental experiment here represents overexpression and to what extent.
3. Along similar lines, miRNA deletion either genetically or with antagomirs has also been used to assess miRNA target hierarchy ("derepression") and could significantly augment the dataset here (for example, by performing the same analysis on bulk cell samples deleted for miR-199 if expressed endogenously).
4. In figure 2 the authors chose to model 4 transcripts targeted by one miRNA. It would be helpful to consider the case of transcript targeted by two miRNAs, such as the 2 seeds overexpressed here. Since combinatorial regulation by miRNA is so common for endogenous genes this comparison could provide a useful context for these results.
5. In figure 2D-F, the analysis should include the non-targeted "nt" set of genes from figure 1E. These would not be expected to change C_v , or mean \log_2 sum target levels, with changes in GFP (miRNA) levels and this comparison is necessary to assess the specificity of the observed changes.
6. Figure 5: The authors do not present sufficient justification for why the ranges of K_m and k_{cat} used are the correct ones for assessing the ceRNA hypothesis. Wouldn't a more straightforward approach to evaluating the ceRNA hypothesis be given in figure 2F, comparing 5p/3p targets to the non-targeted genes? If the latter do not show a similar spike or upward shift in mean or distribution of pearson correlation coefficients this would lend support to the ceRNA idea. Additionally, the data do support a potential ceRNA effect within a limited range. Hence it is confusing why the authors conclude "this mode of regulation should be rare" without modelling a larger parameter space. Finally, the modelling here would be strengthened by comparing the chosen parameters to those empirically observed for the set of miR-199 targets in HEK293, and justification as to why this can be abstracted into a broad principle about the applicability of the ceRNA hypothesis to all miRNA.

Minor points:

1. The introduction contains some confusingly or absolutely worded statements that could be clarified. For example, miRNA function is described in a way that suggests all miRNA first repress translation and then promote target degradation, but many miRNA are thought to function primarily by one mechanism or the other. Additionally, while the authors note many miRNA are "dispensable for development and viability" in *C. elegans*, in fact one of the early foundational discoveries in this field was of let-7, which is central to development in *C. elegans* and mammals. We would encourage the authors to cross reference the introduction with a well written review of microRNA function from a primarily cell biological perspective, such as Bartel D, Cell 2008.
2. Axis labelling of figures 1E/1F is unclear: The text and legend refer to GFP hi vs low, so what is $\log_2(\text{ind}/\text{ctrl})$? Similarly the y-axis of the inset in figure 1E should be labelled.
3. The second cell line i199-KTN1 does not really add much here and we might suggest moving it to the supplement. The reversal of 5p/3p ordering between the two cell lines in figure 2E vs 2H, for example, could be confusing without impacting the authors central points.
4. The data in 2B and 2C is not sufficiently commented upon, for example why does target c show such a large increase in C_v compared to the others and why does r change with higher amplitude for pairings lacking c. I could reason it out after thinking about it for awhile but it would be better to

explain directly in the text.

5. I found it personally difficult to understand what was being assessed in Figure 3C and what this represented. Perhaps it could be further explained or presented in a different way to make it more accessible to the cell biological community

6. In Figure 4F, the x-axis is labelled as Nr of targets but is referred to in the text as prediction rank. This is confusing labelling and should be clarified as to what is actually plotted here. It may be worthwhile to separate the 3 different comparisons shown together on different graphs if the axes represent different quantities.

Reviewer #3:

The authors present a study that gives new insights into miRNA function, an exciting and in many aspects novel topic. By incrementally overexpressing a single miRNA and profiling the resulting transcriptome-wide changes in single cells, they provide evidence that the miRNA can influence the expression noise and correlation patterns of its targets. They further present established mathematical models for miRNA function, apply these to simulate data, and evaluate the concordance with the experimental data. The study is well conducted, interesting and potentially important. However, some key analyses must be clarified.

Major comments:

1. The subfigures 2E-F and 2H-I are essential to the biological insights of the study. However, the methods used to normalize the data are not completely clear, and in particular it is not clear if these methods are robust. The authors write that the coefficient of variation (CV) and rho values are normalized to the values of 300 random mRNAs that have expression and downregulation similar to the targets, but that are not predicted miRNA targets. First, it is not clear why control transcripts were chosen that are downregulated. This would imply that these transcripts are also regulated when the miRNA is overexpressed, possibly by secondary effects. So it seems that here the effect of miRNA repression is compared to other, unknown, repression mechanisms, when it would seem more logical to compare to unperturbed transcripts. Second, it is not clear if the set of 300 random control transcripts was selected only once, or if the analysis was repeated several times to test the robustness of the results. In summary, the current subfigure 2E-F and 2H-I should be complemented by four new subfigures, where the entire set of (miR-199a) non-target transcripts are considered, and the targets and non-targets are both plotted in the same subfigures, using distinct colors. It is clear that the non-targets may not have the same expression values as the targets, which may influence the CV values. However, the CV values can be normalized to be relatively independent from the expression values, for instance using the approach from Faure et al., *Cell Systems*, 2017.

2. Related to this, it is known that single-cell sequencing introduces large amounts of technical noise, in part because of the relatively low sensitivity in the detection of the individual molecule. The authors should discuss how this influences their results, in particular the measured CV values. If technical spike-in sequences were used, these can be analyzed to estimate technical versus biological noise.

Minor comments:

1. The results from the Bornkamm et al. study suggests that expression from the bi-directional promoter may not be exactly the same in the two directions, for a given cell. This is not a major problem, but the authors could discuss it in a sentence or two. I assume that the authors have already attempted to detect the (likely cleaved) pri-miRNAs in their single-cell sequencing data. It would of course be interesting to correlate such expression values with those of the GFP transcript.

2. In subfigures 2E and 2F, the normalized CV values appear to behave erratic for values below a log₂ GFP values of 4. In particular, the behavior seems different for the two cells. The reason for this should be explained and discussed in a few sentences.

Response to reviewers' comments**Reviewer #1:**

Rzepiela et al. propose a simple M-K kinetic model for miRNA regulation of multiple targets and fit this model to results from biochemical assays generated for this manuscript. Biochemical assays include the upregulation of has-miR199-a in 293 cells and a sister cell type overexpressing the 3'UTR of KTN1, following by scRNA-Seq. Based on their model and experiment, the authors conclude that (1) they have obtained the first response parameters for miRNA targets, and (2) that miRNA correlate target expression and increase the variability of target expression across cells. I find limited novelty in the finding of this manuscript and don't accept the claims for originality, as presented by the authors. In my view, the manuscript's novelty and originality are insufficient for publication at MSB.

Major issues are listed below

1. The authors neglected to cite previous work on the topic, including (Arvey et al., 2010; hyun Kim et al., 2013; Luna et al., 2015). In particular, both there has been a slew of proposed kinetic models for miRNA regulation that are similar in nature to what was proposed here (Chen et al. 2016, Vasilescu et al. 2016, Cesana et al. 2013, Yuan et al. 2015, Ala et al. 2013, Bosia et al. 2013, and others...). Some, including Luna et al. and Bosson et al., had experimental data that could be used to fit these models. The conclusions here are mostly in line with their results.

We disagree with this reviewer's summary of our work. Its novelty is not in the kinetic model of miRNA regulation. It is rather in combining single cell sequencing data with the inference method that we developed, to infer parameters of miRNA target interactions such as K_m and 'free Ago critical' in parallel, for all miRNA targets, expressed in their native context. To our knowledge, this design is unique in the literature. It allows us to investigate the response of the entire target network of targets, in particular questions about target competition, in the natural expression context, not upon extremely high induction of a transgene. We think that this is an important distinction between our system and others in which competition between targets has been investigated with mathematical models, including those which the reviewer mentions (Luna et al. and Bosson et al.).

We are aware that many kinetic models have been proposed and investigated, primarily computationally. We indeed cite in the introduction several theoretical and experimental publications (Bosia et al. 2015, Figluizzi et al. 2013, Denzler et al. 2014, Bosson et al. 2014) that most directly relate to and motivated our work. We also make it clear that *"We obtained experimental evidence for behaviors that were previously **suggested by computational models or evaluated only with miRNA target reporters**. These include the non-linear, ultra-sensitive response of miRNA targets to changes in the miRNA concentration as well as the dependency of the variability in target levels between cells on the concentration of the miRNA."*

The reviewer suggests that the Bosson et al. 2014 and Luna et al. 2015 papers provide data to which kinetic models as the one we use here can be fitted. Here again, we emphasize that our aim was to infer the parameters of interaction for the entire population of targets of a miRNA in their endogenous expression context. This cannot be done based on the experimental data in the cited papers. There, different types of data such as reporter gene expression levels and CLIP sequencing reads were used to test specific hypotheses about ceRNA effects and miRNA sequestration. Where similar questions were asked, the conclusions in those studies were generally in line with ours, as the reviewer mentions. However, data from our unique experimental design allows us to investigate other aspects as well, such as the miRNA target hierarchy and how the entire network of miRNA targets, expressed from their own promoters, responds to miRNA induction.

Finally, in the discussion section of Luna et al. single cell analyses mentioned as an important direction for future studies of miRNA sponging: ‘*Combined with HCV genotype, dynamic replication variation within the liver, and host variability in innate immune responses (Sheahan et al, 2014), a complex picture of HCV infection emerges that would largely mask observations of HCV sponge effects in bulk cell or tissue AGO-CLIP measurements. As it remains possible that functional effects of such a sponge may impact highly infected cells, our data highlight the possibility of searching for transcriptome level changes to the miR-122 target network in response to HCV infection in individual cells.*’ Our work provides a framework for such studies, which can only become more informative as the depth and accuracy of single cell gene expression measurements increases.

2. While the authors address the issue of multiple targets per miRNA, they do not address issues that arise from targets sharing multiple miRNAs (Chiu et al., 2015). Can the model be extended to account for both multiple targets and regulators? Without this, conclusions about the expected frequency of ceRNA regulation are suspect.

We do not understand what the reviewer finds ‘suspect’ in our conclusions. The ceRNA hypothesis does not require that targets share multiple RNAs. In fact, one of the most compelling cases of miRNA sponging by an individual target corresponds to the Cdr1as circular RNA, which has a very large number of miR-7 binding sites along with a single miR-671 binding site, yet has a clear impact on miR-7 targets (Memczak et al, 2013). To the extent to which the results that we obtained here for two distinct miRNAs apply to other miRNAs, the impact of overexpressing a target on the expression of other targets should be small, except under conditions in which the overexpressed target accounts for a large fraction of the target population of each individual miRNA. Even if one target shares more than one miRNA with others, the number of miRNA molecules (even of distinct types) sponged by the overexpressed target is a small proportion of entire miRNA population.

3. The authors are studying cells in culture. Can the authors argue for the benefit of using scRNAs that produce fewer reads per cells in this homogeneous system? I find the idea that perturbations lead to variability across cells to be both natural and unsurprising.

We do not understand what the reviewer means with “the benefit of using scRNAs that produce fewer reads per cells in this homogeneous system”. Furthermore, we did not suggest that the observed variability across single cells is not natural or surprising. We made use of this variability to obtain a large number of data points of target expression over a wide miRNA expression range.

4. On a minor note, the authors elected to study 300 targets predicted by their program. How do these compare to responses by other genes?

The response of non-targets is shown in Figure 1. With respect to these, the 300 targets (a relatively large number) undergo significant down-regulation upon miRNA induction, as expected.

References

- Arvey, A., Larsson, E., Sander, C., Leslie, C.S., and Marks, D.S. (2010). Target mRNA abundance dilutes microRNA and siRNA activity. *Molecular systems biology* 6, 363.
- Chiu, H.S., Llobet-Navas, D., Yang, X., Chung, W.J., Ambesi-Impiombato, A., Iyer, A., Kim, H.R., Seviour, E.G., Luo, Z., Sehgal, V., et al. (2015). Cupid: simultaneous reconstruction of microRNA-target and ceRNA networks. *Genome Res* 25, 257-267.

hyun Kim, D., Grün, D., and van Oudenaarden, A. (2013). Dampening of expression oscillations by synchronous regulation of a microRNA and its target. *Nature genetics* 45, 1337.

Luna, J.M., Scheel, T.K., Danino, T., Shaw, K.S., Mele, A., Fak, J.J., Nishiuchi, E., Takacs, C.N., Catanese, M.T., de Jong, Y.P., et al. (2015). Hepatitis C Virus RNA Functionally Sequesters miR-122. *Cell* 160, 1099-1110.

Reviewer #2:

In "Single cell mRNA profiling reveals the hierarchical response of miRNA targets to miRNA induction," the authors utilize both experimental and modelling approaches to assess the impact of miRNA on mRNA levels. The primary experimental approach is the introduction of an inducible bidirectional vector expressing both GFP and pri-miRNA (hsa-miR-199a-5p/3p) into HEK 293 cells, with measurement of GFP mRNA then serving as a surrogate for miRNA expression. The authors then assess the HEK 293 cells both by bulk sequencing and single cell sequencing at different levels of GFP (miRNA) expression and assess the impact on predicted miR-199 targets. This experimental approach is complemented by modelling considering the dynamics of free mRNA and Ago/miRNA bound mRNAs in various contexts.

The work attempts to address several aspects of miRNA biology, including the previous observations that only a subset of predicted mRNA targets are observed to change upon miRNA overexpression and the somewhat conflicting viewpoints surrounding the idea that mRNAs compete for Ago-miRNA binding such that changes in levels of an endogenous gene can influence the totality of bound targets (ceRNA hypothesis). The authors generate a dataset in which miRNA levels vary across a range of expression within a cell population and then use single cell mRNA analysis to determine the impact on targets. To our knowledge this dataset is novel and of interest to the greater miRNA community (and likely the modelling/inference community as well). The authors then apply various modelling techniques to define two parameters K_m and ACF that define the sensitivity of a particular mRNA to miRNA (in this case hsa-199) and argue the range of observed and predicted parameters defines a hierarchy of miRNA-mRNA sites. While the assertion that miRNA sites exist in a hierarchy of occupancy or activity is not new, their inference of this hierarchy from empirical single cell sequencing data represents an interesting approach. Where this work falls a bit short in its current form is in its failure to link the observed and inferred hierarchy here with biochemical features of miRNA-mRNA interactions, such as the idea that binding site type (6- vs 7- vs 8-mer) or other such features (DG and/or thermal stability of interaction) might dictate Ago occupancy.

We appreciate the reviewer's accurate summary of our work, as well as for pointing out the novelty of our study design.

Major points:

1. As mentioned, the hierarchy of sites is not related to any features of miRNA-mRNA interaction beyond the MIRZA-G-C score. This would greatly add to broadening the interest of the findings. Additionally, it is unclear why the authors chose this ranking algorithm as opposed to TargetScan or biochemically defined miRNA-mRNA interactions

(for example through cross linking immunoprecipitation experiments, if this data is available for HEK293).

We used the MIRZA-G-C algorithm that we developed and showed that its performance is comparable to that of TargetScan, yet using a much smaller set of parameters (Gumienny & Zavolan, 2015). To address reviewer's question, we now provide in the Appendix a parallel analysis carried out based on TargetScan-predicted targets (version 6.2, which also provides individual components of the score) predictions. Note that MIRZA-G-C reports features of individual sites and the probabilities of sites being functional in miRNA repression (positive quantity), while TargetScan reports contributions of features to the log-fold-change of targets (which should be a negative quantity). Thus, similar features that correlate with target responses should give correlations of opposite sign. Both MIRZA-G-C and TargetScan are trained to predict mRNA level changes upon miRNA transfection, and indeed, the scores that they give correlate best (in absolute value) with the fold-change of the predicted targets in cells with high miRNA expression compared with low miRNA expression. However, some structure in the contribution of individual features can be glanced. Measures related to the A/U content in the vicinity of sites and their relative location in 3' UTRs are correlated with the FAC and Km, whereas the degree of evolutionary conservation is most informative for the fold-change. We have added the corresponding figure for MIRZA-G-C features as a panel of Figure S5, while the analysis based on TargetScan-predicted targets is in the Appendix.

2. Is there endogenous miR-199 in these cells, and if so what are its levels? The authors need to address whether the fundamental experiment here represents overexpression and to what extent.

In previous work we have shown that miR-199a-3p is undetectable by northern blot (Hausser et al. *Molecular Systems Biology* 9:711 (2013)). Additionally, Figure 1B shows that endogenous miR-199a-5p expression is very low in uninduced cells (a few molecules per cell), reaching $\sim 10^4$ molecules per cell in strongly induced cells, and Figure S1A shows that miR-199a-3p and miR-199a-5p are induced in parallel by doxycycline. To answer even more directly reviewer's question about the degree of miRNA overexpression, we now provide Ago2-CLIP data from cells in which miRNA expression was fully induced with 1 $\mu\text{g/ml}$ doxycycline. We found that miR-199a-3p and miR-199a-5p were among the most abundant miRNAs interacting with Ago2, their variants amounting to 28% of the miRNA population in replicate 1 of the Ago2-CLIP and 9% in replicate 2. Thus, miR-199a-5p and miR-199a-3p are induced to the level of highly expressed endogenous miRNAs, but do not dominate the population of miRNAs. We present the results in new panel B of Figure S1. Finally, Panels E and F of Figure 1 further show that miR-199a-5p/3p targets are downregulated up to ~ 2 -fold in fully induced cells, which is the range of downregulation of miRNA targets in miRNA transfection experiments.

3. Along similar lines, miRNA deletion either genetically or with antagomirs has also been used to assess miRNA target hierarchy ("derepression") and could significantly augment the dataset here (for example, by performing the same analysis on bulk cell samples deleted for miR-199 if expressed endogenously).

We fully agree with the reviewer that the complementary experiment, progressive removal of an abundant miRNA would be very interesting to do. miR-199 is not detectable in parental HEK293 cells, and testing the derepression hierarchy would probably require including another construct, that would allow inducible expression of a miR-199 antimir (and an associated proxy mRNA), to progressively reduce miR-199 expression after full induction. This would be a challenging experiment that we could not address here. Nevertheless, the reviewer may appreciate that we tried to address questions of both robustness and generality by using two closely related, but not identical cell lines, in which the population of putative miR-199 targets of miR-199 is very similar, and the

hierarchy is also expected to be similar. Indeed, we found that the parameters that we inferred for individual targets from the two cell lines are well correlated, in spite of the large noise inherent in single cell sequencing experiments (Fig 4D-E). Furthermore, we took advantage of the fact that both arms of pre-miR-199 give rise to two distinct mature miRNAs in our cell lines, allowing us to demonstrate similar behavior of targets for two distinct miRNAs. The mathematical framework for inferring the K_m s and FAC s is general and we do hope that the community will use our approach on other systems as well, especially as the depth and accuracy of single cell RNA sequencing increases. Efforts to sequence both miRNAs and mRNAs from individual cells are ongoing and this could further increase the applicability and accuracy of our method, as one would not need to use a proxy of miRNA levels.

4. In figure 2 the authors chose to model 4 transcripts targeted by one miRNA. It would be helpful to consider the case of transcript targeted by two miRNAs, such as the 2 seeds overexpressed here. Since combinatorial regulation by miRNA is so common for endogenous genes this comparison could provide a useful context for these results.

Indeed, most genes are probably targeted by multiple miRNAs, and there is ample evidence in the literature that the degree of target repression increases with the number of targeting miRNAs (but also more generally with the quality and number of sites for individual miRNA). There is also a number of theoretical studies of multiple miRNA systems (e.g. (Bosia *et al.*, 2013)) The behavior can be complex, depending on the relative interaction affinities and concentrations of the targeting miRNAs. As the miRNAs induced in our system have relatively little (6 of the top 100 targets and 179 of the 1200-1300 full sets of targets), we stayed closer to the experimental data with the simulations as well.

5. In figure 2D-F, the analysis should include the non-targeted "nt" set of genes from figure 1E. These would not be expected to change C_v , or mean \log_2 sum target levels, with changes in GFP (miRNA) levels and this comparison is necessary to assess the specificity of the observed changes.

In fact, the figure already includes the comparison with non-targets, as we show the ratio of CV and the correlation of \log_2 target levels of targets relative to non-targets that have similar overall expression and fold change upon strong miRNA induction. The reason is that we observed systematic effects in the single cell data (shown in new supplementary figure 3) and we, as the reviewer, were interested in the specificity of the effects for targets. In response to this and reviewer #3 comments, we have revised the figure to compare targets with all non-target genes (not only those with similar expression patterns) and to also quantify the variability of the estimates presented in the figure.

6. Figure 5: The authors do not present sufficient justification for why the ranges of K_m and k_{cat} used are the correct ones for assessing the ceRNA hypothesis. Wouldn't a more straightforward approach to evaluating the ceRNA hypothesis be given in figure 2F, comparing 5p/3p targets to the non-targeted genes? If the latter do not show a similar spike or upward shift in mean or distribution of Pearson correlation coefficients this would lend support to the ceRNA idea.

Although we have explored a wide range of K_m and K_{cat} values, 'interesting' behaviors occur for a relatively small range of parameters, which we tried to capture in our figure. We only focused on targets, because non-targets will by definition not respond to changes in the miRNA concentration in the model, and thus their stochastic expression changes would not be correlated.

Additionally, the data do support a potential ceRNA effect within a limited range. Hence it is confusing why the authors conclude “this mode of regulation should be rare” without modelling a larger parameter space.

We apologize for the confusion of our formulation. We did explore a wide range of parameters, and we clarified this in the revision. Given the number of parameters that can in principle vary, it is difficult to come up with a comprehensive visualization. We chose to illustrate a few sections of the parameter space where a putative ceRNA would have some effect. We also clarified in the revision why we think that ceRNA-based regulation should be rare.

Finally, the modelling here would be strengthened by comparing the chosen parameters to those empirically observed for the set of miR-199 targets in HEK293, and justification as to why this can be abstracted into a broad principle about the applicability of the ceRNA hypothesis to all miRNA.

Some of the parameters that we used in these simulations (K_{ms} , fold changes, target expression levels) were directly sampled from the distributions that we inferred from the studied experimental data, while others (e.g. decay rates, mean k_{on} and k_{off} values) were set to values that we previously inferred from other experimental data sets (see also (Hausser & Zavolan, 2014; Hausser *et al*, 2013)). We note that the information about ranges of k_{on} and k_{off} values is very limited. We thank the reviewer for pointing out that this was not clear in the text, and we hope that this has become much clearer in the revision. We have also expanded our discussion to justify our conclusion regarding the prevalence of ceRNA interactions.

Minor points:

1. The introduction contains some confusingly or absolutely worded statements that could be clarified. For example, miRNA function is described in a way that suggests all miRNA first repress translation and then promote target degradation, but many miRNA are thought to function primarily by one mechanism or the other. Additionally, while the authors note many miRNA are “dispensable for development and viability” in *C. elegans*, in fact one of the early foundational discoveries in this field was of *let-7*, which is central to development in *C. elegans* and mammals. We would encourage the authors to cross reference the introduction with a well written review of microRNA function from a primarily cell biological perspective, such as Bartel D, Cell 2008.

We thank the reviewer for pointing out the lack of clarity. We have emphasized in the revised introduction that we only discuss mammalian miRNAs. For these, we believe that the prevalent paradigm is in fact summarized in a paper from the Bartel group (Eichhorn *et al*, 2014), which explicitly states that the dominant effect of miRNAs is mRNA destabilization, and that translational repression, if it does occur, is an early and transient effect. We have added the reference suggested by the reviewer and we also explicitly mention the developmental effects of *let-7*.

2. Axis labelling of figures 1E/1F is unclear: The text and legend refer to GFP hi vs low, so what is $\log_2(\text{ind}/\text{ctrl})$? Similarly the y-axis of the inset in figure 1E should be labelled.

We revised the text to make it fully consistent with the axis labels (which we have not changed to not clutter the figures).

3. The second cell line i199-KTN1 does not really add much here and we might suggest moving it to the supplement. The reversal of 5p/3p ordering between the two cell lines in figure 2E vs 2H, for example, could be confusing without impacting the authors central points.

We have revised the figure in response to the reviewers' comments. The panels have become even more consistent and, because we think it is important to demonstrate the robustness of the results, we opted to keep these panels in the main text.

4. The data in 2B and 2C is not sufficiently commented upon, for example why does target *c* show such a large increase in *Cv* compared to the others and why does *r* change with higher amplitude for pairings lacking *c*. I could reason it out after thinking about it for awhile but it would be better to explain directly in the text.

We thank the reviewer for this suggestion. We have added a more detailed description of these figure panels in the text.

5. I found it personally difficult to understand what was being assessed in Figure 3C and what this represented. Perhaps it could be further explained or presented in a different way to make it more accessible to the cell biological community

Again, we thank the reviewer for the suggestion, we have also expanded the explanation of these figure panels.

6. In Figure 4F, the x-axis is labelled as Nr of targets but is referred to in the text as prediction rank. This is confusing labelling and should be clarified as to what is actually plotted here. It may be worthwhile to separate the 3 different comparisons shown together on different graphs if the axes represent different quantities.

What is shown is the number of top targets considered. We have updated the figure legend.

Reviewer #3:

The authors present a study that gives new insights into miRNA function, an exciting and in many aspects novel topic. By incrementally overexpressing a single miRNA and profiling the resulting transcriptome-wide changes in single cells, they provide evidence that the miRNA can influence the expression noise and correlation patterns of its targets. They further present established mathematical models for miRNA function, apply these to simulate data, and evaluate the concordance with the experimental data. The study is well conducted, interesting and potentially important. However, some key analyses must be clarified.

We thank the reviewer for his/her positive comments.

Major comments:

1. The subfigures 2E-F and 2H-I are essential to the biological insights of the study. However, the methods used to normalize the data are not completely clear, and in particular it is not clear if these methods are robust. The authors write that the coefficient of variation (CV) and rho values are normalized to the values of 300 random mRNAs that have expression and downregulation similar to the targets, but that are not predicted miRNA targets. First, it is not clear why control transcripts were chosen that are downregulated.

This would imply that these transcripts are also regulated when the miRNA is overexpressed, possibly by secondary effects. So it seems that here the effect of miRNA repression is compared to other, unknown, repression mechanisms, when it would seem more logical to compare to unperturbed transcripts.

This question relates to a question of reviewer #2. The reason we have used this type of normalization here is because we have observed systematic effects on non-targets as well, and thus to evaluate the specificity of the noise or correlation response we compared predicted targets with non-targets. Furthermore, to avoid dependencies of these variables on the expression levels of the transcripts, we have chosen control transcripts that are as close as possible to the predicted targets both in expression level and expression change. In our revised manuscript we now use broadly sampled set or all non targets as control rather than a small selected subset of non-targets. The results are similar. Supplementary figure 3 now shows that even when using a normalized variance (pagoda method of Fan et al., Nat.Meth. 2016), targets have higher variation than non-targets, and that in spite of the systematic effects on pairwise correlations of gene expression as a function of GFP expression, targets are more strongly correlated in the region where they respond most sensitively to the miRNA.

Second, it is not clear if the set of 300 random control transcripts was selected only once, or if the analysis was repeated several times to test the robustness of the results.

Indeed, we repeated this selection multiple times (correlations) or used all genes (C_V) to and included error bars to the figures.

In summary, the current subfigure 2E-F and 2H-I should be complemented by four new subfigures, where the entire set of (miR-199a) non-target transcripts are considered, and the targets and non-targets are both plotted in the same subfigures, using distinct colors. It is clear that the non-targets may not have the same expression values as the targets, which may influence the CV values. However, the CV values can be normalized to be relatively independent from the expression values, for instance using the approach from Faure et al., Cell Systems, 2017.

We have indeed applied the approach suggested by the reviewers and show the corresponding results in the updated panels of Figure 2 and Supplementary Figure 3 (see also response to the previous comment).

2. Related to this, it is known that single-cell sequencing introduces large amounts of technical noise, in part because of the relatively low sensitivity in the detection of the individual molecule. The authors should discuss how this influences their results, in particular the measured CV values. If technical spike-in sequences were used, these can be analyzed to estimate technical versus biological noise.

The reviewer is entirely right that the sensitivity of single cell sequencing is still rather low. To generate our data, we used a 10x genomics platform which allows us to obtain a large number of cells, but for which the sample preparation protocol does not include spike-ins. Nevertheless, the downregulation of target levels, whether computed from mRNA levels inferred from single cells or from bulk sequencing are very similar, indicating that our data processing is reasonable. We have expanded our discussion to cover this aspect. We also thank the reviewer for suggesting the normalization method, which we have now applied, obtaining similar results. Finally, we also assessed the robustness of our results in other ways, such as through the use of two distinct but closely related cell lines and also by analyzing two distinct miRNAs.

Minor comments:

1. The results from the Bornkamm et al. study suggests that expression from the bi-directional promoter may not be exactly the same in the two directions, for a given cell. This is not a major problem, but the authors could discuss it in a sentence or two. I assume that the authors have already attempted to detect the (likely cleaved) pri-miRNAs in their single-cell sequencing data. It would of course be interesting to correlate such expression values with those of the GFP transcript.

We expanded our discussion of the construct. Indeed, it would have been ideal to measure the expression of the miRNA directly, and the pre-miRNA may have been a good second option, but unfortunately, the poly(A) selection that is inherent to our single cell sequencing protocol would not allow us to capture the pre-miRNA. As an aside, we have also tried to detect it in an earlier experiment, that was done with much smaller cell numbers on a Fluidigm instrument, and we were also unsuccessful. This is perhaps not surprising, because the pre-miRNA expression is generally quite low.

2. In subfigures 2E and 2F, the normalized CV values appear to behave erratic for values below a log₂ GFP values of 4. In particular, the behavior seems different for the two cells. The reason for this should be explained and discussed in a few sentences.

At the reviewers' suggestions, we have updated these figures, particularly using all genes as background. Although the noise remains large, differences between cell lines are much smaller, and the effect of miRNAs on the CV of their targets is apparent over the entire range of miRNA expression. We have also added a few sentences about this in the description of the figure.

Bosia C, Pagnani A & Zecchina R (2013) Modelling Competing Endogenous RNA Networks. *PLoS One* **8**: e66609

Eichhorn SW, Guo H, McGeary SE, Rodriguez-Mias RA, Shin C, Baek D, Hsu S-H, Ghoshal K, Villén J & Bartel DP (2014) mRNA destabilization is the dominant effect of mammalian microRNAs by the time substantial repression ensues. *Mol. Cell* **56**: 104–115

Gumienny R & Zavolan M (2015) Accurate transcriptome-wide prediction of microRNA targets and small interfering RNA off-targets with MIRZA-G. *Nucleic Acids Res.* Available at: <http://dx.doi.org/10.1093/nar/gkv050>

Hausser J, Syed AP, Selevsek N, van Nimwegen E, Jaskiewicz L, Aebersold R & Zavolan M (2013) Timescales and bottlenecks in miRNA-dependent gene regulation. *Mol. Syst. Biol.* **9**: 711

Hausser J & Zavolan M (2014) Identification and consequences of miRNA-target interactions - beyond repression of gene expression. *Nat. Rev. Genet.* **15**: 599–612

Memczak S, Jens M, Elefsinioti A, Torti F, Krueger J, Rybak A, Maier L, Mackowiak SD, Gregersen LH, Munschauer M, Loewer A, Ziebold U, Landthaler M, Kocks C, le Noble F & Rajewsky N (2013) Circular RNAs are a large class of animal RNAs with regulatory potency. *Nature* **495**: 333–338

Sheahan T, Imanaka N, Marukian S, Dorner M, Liu P, Ploss A & Rice CM (2014) Interferon lambda alleles predict innate antiviral immune responses and hepatitis C virus permissiveness. *Cell Host Microbe* **15**: 190–202

Thank you for sending us your revised manuscript. We have now heard back from reviewer #3 who was asked to evaluate your study. As you will see below, the reviewer thinks that the study has significantly improved as a result of the performed revisions. S/he lists however two remaining concerns, which we would ask you to address in a minor revision.

Before we formally accept the study, we would also ask you to address the following editorial issues:

REFEREE REPORTS

Reviewer #3:

It is comforting that the analyses shown in Figure 2E-F and 2H-I appear to be robust to the choice of non-target control transcripts and the new Supplementary Figure S3 is very useful. However, a careful comparison of the two figures appears to reveal inconsistencies. As I understand it, for instance Figure 2I is generated by dividing the rp values for hsa-miR-199a-5p and hsa-miR-199a-3p (red and blue lines) with the rp values for the non-target control transcripts (grey line) in Supplementary Figure S3F. Is this correct? If so, it is surprising that in Figure 2I, the normalized rp of hsa-miR-199a-5p (red) is around 2 for log₂ GFP values of 5, while in Figure S3F, the rp of hsa-miR-199a-5p (red) appears to be only slightly above the rp of the control transcripts (grey) for log₂ GFP values of 5. In other words, a slight increase in Figure S3F results in a doubling in Figure 2I. If I am misreading the values in these figures, then please provide high-resolution figures with grid background, for the purpose of reviewing only. If Figure 2I and Figure S3F indeed present different data, then please explain.

On a side note, in the manuscript it is well described how the target set was selected. It would be great to know also exactly how the non-targets were selected (for instance, are these the complete set of human protein coding genes, or are these only the set that have miRNA target sites (for miRNAs other than hsa-miR-199a)?

Response to reviewers' comments

Reviewer #3:

It is comforting that the analyses shown in Figure 2E-F and 2H-I appear to be robust to the choice of non-target control transcripts and the new Supplementary Figure S3 is very useful. However, a careful comparison of the two figures appears to reveal inconsistencies. As I understand it, for instance Figure 2I is generated by dividing the rp values for hsa-miR-199a-5p and hsa-miR-199a-3p (red and blue lines) with the rp values for the non-target control transcripts (grey line) in Supplementary Figure S3F. Is this correct? If so, it is surprising that in Figure 2I, the normalized rp of hsa-miR-199a-5p (red) is around 2 for log₂ GFP values of 5, while in Figure S3F, the rp of hsa-miR-199a-5p (red) appears to be only slightly above the rp of the control transcripts (grey) for log₂ GFP values of 5. In other words, a slight increase in Figure S3F results in a doubling in Figure 2I. If I am misreading the values in these figures, then please provide high-resolution figures with grid background, for the purpose of reviewing only. If Figure 2I and Figure S3F indeed present different data, then please explain.

We thank the reviewer for his/her positive comments. In Figure 2I the r_p values for hsa-miR-199a-5p and hsa-miR-199a-3p (red and blue lines) are indeed divided by the r_p values for the non-target control transcripts (grey line) shown in Appendix Figure S3F (previously called Supplementary Figure S3F). The effect is better visible in the magnified section of Figure S3F, shown below, which focuses on \log_2 GFP range 4-6. The r_p values of red/blue targets are about two-three times higher compared to r_p values of the control genes (gray) for this expression range. This difference around \log_2 GFP levels 4-6 explains the peaks in Figures 2F and 2I.

Although unrelated to the reviewer's question, please note that by mistake, we inverted red and blue colors in Figures 2F and 2I. The corrected Figure 2 is attached as a separate high resolution file in this revision.

On a side note, in the manuscript it is well described how the target set was selected. It would be great to know also exactly how the non-targets were selected (for instance, are these the complete set of human protein coding genes, or are these only the set that have miRNA target sites (for miRNAs other than hsa-miR-199a)?

Non-targets are all genes in the NCBI's Entrez Gene ID space (*homo sapiens*), except of hsa-miR-199a 3p/5p target genes (were MIRZA miRNA target scores were computed for RefSeq transcripts and mapped to corresponding Entrez Gene IDs (Gumienny & Zavolan, 2015)). Note that for all the analysis in the manuscript, as explained in the *Read mapping and data preprocessing* section of the *Methods*, we use genes that are expressed above the threshold 7 TPM in the given data set. In the manuscript we included additional clarification about subset of non-targets in the *Computation of the coefficient of variation of target expression* section of the *Methods*.

Magnified section of Appendix Figure S3F. Mean Pearson pairwise correlation coefficients for miRNA targets (hsa-miR-199a-5p shown in red and hsa-miR-199a-3p shown in blue) targets in function of GFP expression in i199-KTN1 cells. Mean from 50 calculation evaluations of random selection of 100 non target genes is shown as grey line. Means were calculated from the two hundred cells with GFP expression closest to a specific expression level . Standard deviations are shown along mean values.

References

Gumienny R & Zavolan M (2015) Accurate transcriptome-wide prediction of microRNA targets and small interfering RNA off-targets with MIRZA-G. *Nucleic Acids Res.* Available at: <http://dx.doi.org/10.1093/nar/gkv050>

Corresponding Author Name: Mihaela Zavalan
Journal Submitted to: Molecular Systems Biology
Manuscript Number: MSB-18-8266